# Variability in *n*-caprylate and *n*-caproate producing microbiomes in reactors with in-line product extraction

Catherine M. Spirito,[1,2] Timo N. Lucas,[3] Sascha Patz,[3] Byoung Seung Jeon,[4] Jeffrey J. Werner,[5] Lauren H. Trondsen,[1] Juan J. Guzman,[1] Daniel H. Huson,[3] Largus T. Angenent[1,4,6,7,8]

**ABSTRACT** Medium-chain carboxylates (MCCs) are used in various industrial applications. These chemicals are typically extracted from palm oil, which is deemed not sustainable. Recent research has focused on microbial chain elongation using reactors to produce MCCs, such as *n*-caproate (C6) and *n*-caprylate (C8), from organic substrates such as wastes. Even though the production of *n*-caproate is relatively well-characterized, bacteria and metabolic pathways that are responsible for *n*-caprylate production are not. Here, three 5 L reactors with continuous membrane-based liquid-liquid extraction (i.e., pertraction) were fed ethanol and acetate and operated for an operating period of 234 days with different operating conditions. Metagenomic and metaproteomic analyses were employed. *n*-Caprylate production rates and reactor microbiomes differed between reactors even when operated similarly due to differences in $H_2$ and $O_2$ between the reactors. The complete reverse β-oxidation (RBOX) pathway was present and expressed by several bacterial species in the *Clostridia* class. Several *Oscillibacter* spp., including *Oscillibacter valericigenes*, were positively correlated with *n*-caprylate production rates, while *Clostridium kluyveri* was positively correlated with *n*-caproate production. *Pseudoclavibacter caeni*, which is a strictly aerobic bacterium, was abundant across all the operating periods, regardless of *n*-caprylate production rates. This study provides insight into microbiota that are associated with *n*-caprylate production in open-culture reactors and provides ideas for further work.

**IMPORTANCE** Microbial chain elongation pathways in open-culture biotechnology systems can be utilized to convert organic waste and industrial side streams into valuable industrial chemicals. Here, we investigated the microbiota and metabolic pathways that produce medium-chain carboxylates (MCCs), including *n*-caproate (C6) and *n*-caprylate (C8), in reactors with in-line product extraction. Although the reactors in this study were operated similarly, different microbial communities dominated and were responsible for chain elongation. We found that different microbiota were responsible for *n*-caproate or *n*-caprylate production, and this can inform engineers on how to operate the systems better. We also observed which changes in operating conditions steered the production toward and away from *n*-caprylate, but more work is necessary to ascertain a mechanistic understanding that could be predictive. This study provides pertinent research questions for future work.

**KEYWORDS** chain elongation, hydrogen, oxygen, caproate, hexanoate, octanoate, caprylate, medium-chain carboxylate, bacteria microcompartments

Medium-chain carboxylates (MCCs), such as *n*-caproate and *n*-caprylate, are utilized in a variety of industrial and agricultural applications, including as biofuel precursors, anticorrosion agents, plasticizers, personal care products, feed additives, and antimicrobials (1). MCCs are typically produced as a byproduct of palm oil refining (2).

Address correspondence to Largus T. Angenent, l.angenent@uni-tuebingen.de.

Catherine M. Spirito and Timo N. Lucas contributed equally to this article. The author order was determined based on their contribution to the study and article.

L.T.A. has ownership in Capro-X, Inc., which is a start-up company that is commercializing a chain-elongating biotechnology production platform.

See the funding table on p. 16.

Recent research has focused on producing MCCs in open-culture reactors from organic substrates, including wastes, as part of a circular economy. MCCs have a relatively low solubility in water in their undissociated form. Therefore, MCCs can be extracted from aqueous broths via techniques, such as in-line product extraction, to address MCC toxicity issues and to increase volumetric production rates (3, 4). Laboratory studies demonstrated the efficient production of MCCs by anaerobic fermenter microbiomes at rates comparable to methane production by anaerobic digester microbiomes (4, 5). MCCs are produced via pure and open cultures from a variety of substrates, including (1) synthetic substrates utilizing ethanol (3, 6, 7) or lactic acid (8, 9) as the electron donor (2), organic wastes, and (3) industrial side streams (10–18).

MCCs are often produced via the reverse β-oxidation (RBOX) pathway in which ethanol, lactic acid, or another electron donor is oxidized to acetyl-CoA. Short-chain carboxylates, such as acetate and *n*-butyrate, are then chain elongated to longer-chain carboxylates, such as *n*-caproate (six-carbon chain) and *n*-caprylate (eight-carbon chain) (1, 19–21) (Fig. 1). Chain elongation is a cyclic process in which acetyl-CoA enters the cycle and is condensed with an acyl-CoA by acetyl-CoA C-acyltransferase (Thiolase II) (ACAT) to form an acyl-CoA that is two carbon atoms longer than its substrate. The product of this reaction is further reduced by 3-hydroxy-acyl-CoA dehydrogenase (HAD) or 3-hydroxy-butyrl-CoA dehydrogenase (HBD) (Fig. 1). This product is then dehydrated to 2-enoyl-CoA by enoyl-CoA dehydratase (ECH) and further reduced by an acyl-CoA dehydrogenase (ACD) or butyl-CoA dehydrogenase to form an elongated acyl-CoA (Fig. 1). Finally, terminal enzymes acetyl-CoA transferase (CoAT) or thioesterase (TE) act to remove coenzyme A from the terminal acyl-CoA molecule and release the corresponding acid (Fig. 1). Energy is conserved during the RBOX pathway via flavin-based electron bifurcation and the Rnf respiratory complex (RNF) in which an electron-bifurcating acyl-CoA dehydrogenase complex utilizes two electron-transfer flavoproteins (Fig. 1) (22). Prior research primarily focused on RBOX as the pathway for MCC production (4, 7, 23–25). Recent studies have suggested that the fatty acid biosynthesis (FAB) pathway may also play a role (26, 27). However, it should be noted that all bacteria use the anabolic FAB pathway to build their phospholipid membranes. In addition, FAB is an anabolic process, it consumes energy rather than producing energy that is necessary for bacterial growth in anaerobic conditions without inorganic electron acceptors for respiration.

Previously, both pure- and open-culture studies identified multiple bacterial strains that produce *n*-caproate. Known *n*-caproate-producing bacteria primarily belong to the phylum Firmicutes, except for *Rhodospirillum rubrum* (28). Within the phylum Firmicutes, *n*-caproate-producing bacterial strains have been isolated and identified from six genera: *Caproicibacter*, *Caproiciproducens*, *Clostridium* (29–31), *Eubacterium* (32, 33), *Megasphaera* (34), and *Pseudoramibacter* (24). In open-culture reactor studies, certain bacteria have been associated with high *n*-caprylate production rates, including *Burkholderia* spp., *Clostridium* group IV spp., *Desulfosporosinus meridiei*, *Oscillospira* spp., Rhodocyclaceae K82 spp., unknown Ruminococcaceae, and *Sphingobacterium multiform* (3, 4, 11). These studies were based on 16S rRNA gene sequencing data. Few bacterial isolates have been shown to produce *n*-caprylate. This is attributed to the microbial toxicity of *n*-caprylate and the lack of measurement of *n*-caprylate in prior studies. In 1967, *Ramibacterium alactolyticum*, renamed *Pseudoramibacter alactolyticus*, was shown to produce *n*-caproate and *n*-caprylate from glucose (35). A previous pure-culture reactor study observed the production of relatively low concentrations of *n*-caprylate by *Clostridium kluyveri*, which is a well-known chain-elongating bacterium, in a reactor fed a 10:1 molar ratio of ethanol and acetate (i.e., syngas effluent), operated at pH 7 and with an in-line membrane-based liquid/liquid extraction (i.e., pertraction) system to reduce the toxicity (10). At lower pH levels, a lower rate of *n*-caprylate production was observed. For the open-culture studies, a shotgun metagenomics study found an uncultured Clostridiales order bacterium, *Candidatus Weimeria bifida*, gen. nov., sp. nov., which could produce *n*-caprylate from xylose (23, 24). Research is needed to understand essential players and metabolic pathways in these reactors that optimize *n*-caprylate production.

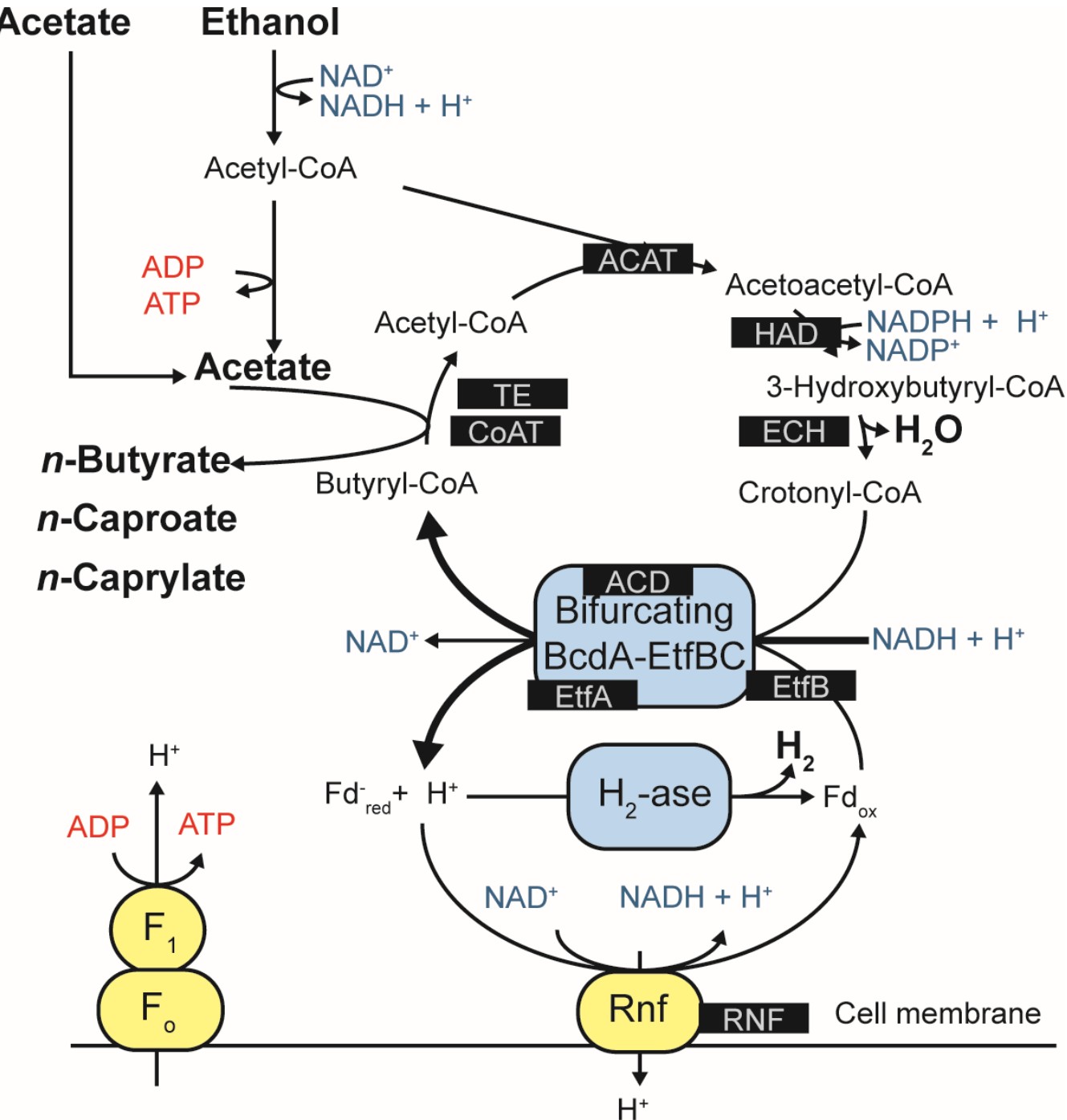

**FIG 1** The RBOX pathway investigated in this study. The enzymes we examined in this study are highlighted in black boxes. The figure was modified with permission from Angenent et al. (21). RBOX pathway enzymes are ACAT, acetyl-CoA C-acyltransferase (Thiolase II); HAD, 3-hydroxy-acyl-CoA dehydrogenase; ECH, enoyl-CoA dehydratase; ACD, acyl-CoA dehydrogenase; EtfA/B, electron-transfer-flavoprotein subunit A/B; CoAT, acetyl CoA-transferase; TE, thioesterase; RNF, Rnf respiratory complex.

Here, we investigated the role of the RBOX pathway in producing *n*-caproate and *n*-caprylate. Our original objective was to build and operate three equal stain-less steel reactor systems to prevent O$_2$ intrusion as much as possible. As an independent operating unit, we planned different H$_2$ concentrations for each system. However, we were unsuccessful in: (i) preventing O$_2$ intrusion; and (ii) utilizing H$_2$ as an independent parameter. Regardless, we obtained pertinent data by operating the three

5 L open-culture reactors with in-line product extraction throughout 234 days. We employed Illumina 16S rRNA gene sequencing, shotgun metagenomics, and metaproteomics to characterize microbiomes. The reactors were fed ethanol and acetate and produced *n*-caproate and *n*-caprylate. Even though the reactors were all provided the same substrates and produced MCCs, their microbiota differed. Several bacterial species belonging to the class Clostridia, including *Oscillibacter valericigenes*, expressed the majority of the RBOX pathway. *Oscillibacter* spp. members were found to be positively correlated with *n*-caprylate production rates. The aerobic bacterium *Pseudoclavibacter caeni* was one of the abundant bacteria in the reactor samples, regardless of *n*-caprylate production rates. *P. caeni* may have acted as an $O_2$ scavenger in the system or provided other unknown roles for producing *n*-caprylate.

## RESULTS

### The performance of the three reactors differed despite similar operating conditions

We operated three stainless-steel, continuously stirred reactors with a 5 L working volume and in-line product extraction at mesophilic conditions and a pH of 5.5 (Table 1; Fig. S1). The effluent carboxylate and ethanol concentrations during the 75-day reactor startup period can be found in Fig. S2. After the 75-day start-up period and at the start of Period I, we mixed the microbiota from all three reactors and then operated the three reactors similarly throughout Period I (without sparging). Regardless, the performance of the three reactors was not similar and varied during this period. Reactors 1 and 2 achieved promising and similar overall medium-chain production rates (Fig. 2), but Reactor 3 performed poorly during Period 1 with a lower *n*-caprylate production rate compared to Reactors 1 and 2 (Fig. 2). Reactor 1 exhibited a higher selectivity (i.e., wanted products compared to the substrate) for *n*-caprylate production compared to Reactor 2. The maximum average volumetric *n*-caprylate production rate was $1.1 \times 10^2 \pm 7.1$ mmol C $L^{-1}$ $d^{-1}$ ($0.080 \pm 0.005$ g $L^{-1}$ $h^{-1}$) during Period 1D for Reactor 1 (Fig. 2B). Small differences in operating conditions, potentially due to differences in the tightness of the reactor seals resulting in different $H_2$ and $O_2$ exchange conditions, seem to have had an amplified impact on chain elongation. This was different from the anaerobic digestion of animal waste, for which we found that four similar reactor operating conditions resulted in almost identical performances after an operating period of 1 year (36).

Our results show that the $H_2$ partial pressure is a sensitive parameter to the *n*-caprylate performance, amplifying minor differences in operating conditions. During Period 1, gas in the headspace of Reactor 3 contained $31\% \pm 9.6\%$ $H_2$ (by volume), whereas $H_2$ was $9.9\% \pm 5.2\%$ and $1.8\% \pm 1.9\%$ of total gas for Reactors 1 and 2, respectively (Table S1). The reducing equivalents $Fd_{red}$ and NADH produced by the RBOX pathway

**TABLE 1** Operating data for three reactors[a]

| Reactor | Period | Days | HRT (d) | OLR (mmol C $L^{-1}$ $d^{-1}$) | Gas flow rate (L $d^{-1}$) | Sparged with |
|---|---|---|---|---|---|---|
| R1 | 1 | 75 to 142 | $8.9 \pm 0.3$ | $1.4 \times 10^2 \pm 7.9$ | 0 | No gas |
| | 2 | 143 to 184 | $9.6 \pm 0.3$ | $1.2 \times 10^2 \pm 6.7$ | $3.38 \pm 4.86$ | $N_2$ off/on |
| | 3 | 185 to 234 | $9.0 \pm 0.3$ | $1.4 \times 10^2 \pm 6.5$ | $24.7 \pm 13.7$ | $N_2$ |
| R2 | 1 | 75 to 142 | $8.9 \pm 0.5$ | $1.4 \times 10^2 \pm 8.4$ | 0 | No gas |
| | 2 | 143 to 184 | $9.0 \pm 0.2$ | $1.3 \times 10^2 \pm 8.8$ | $2.45 \pm 3.53$ | $N_2$ and $H_2$ off/on |
| | 3 | 185 to 234 | $8.9 \pm 0.3$ | $1.4 \times 10^2 \pm 8.2$ | $6.14 \pm 7.00$ | $N_2$, $H_2$ |
| R3 | 1 | 75 to 142 | $7.8 \pm 0.3$ | $1.6 \times 10^2 \pm 7.6$ | $0.50 \pm 0.31$ | No gas |
| | 2 | 143 to 184 | $8.4 \pm 0.5$ | $1.3 \times 10^2 \pm 13$ | $5.02 \pm 5.64$ | $N_2$ off/on |
| | 3 | 185 to 234 | $7.0 \pm 0.6$ | $1.8 \times 10^2 \pm 15$ | $10.2 \pm 4.83$ | $N_2$ |

[a]We report the hydraulic retention time (HRT), organic loading rate (OLR), and gas flow rate for each reactor and each period. The gas flow rate was measured at the outlet of each reactor system. Different gases were utilized as indicated. During Period 2, gas sparging was periodically on and off to the reactors, whereas it was on all the time during Period 3. Mean ± s.e. is reported. R1–3 are Reactors 1–3.

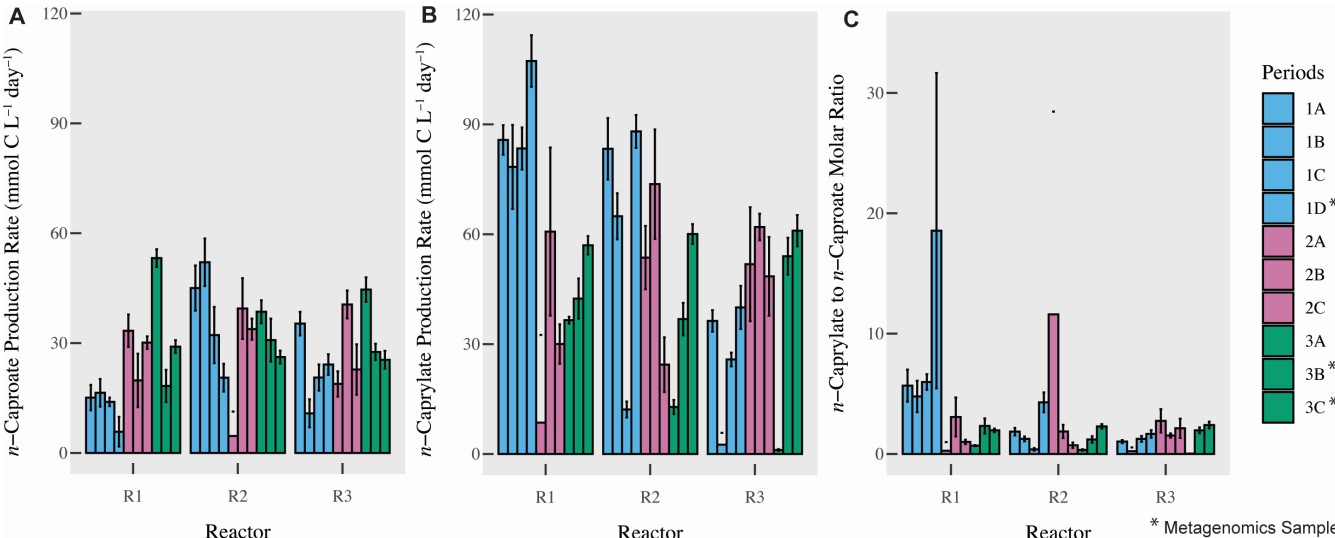

**FIG 2** *n*-Caproate (A) and *n*-caprylate (B) total production rates (mmol C L$^{-1}$ d$^{-1}$) and the molar ratio of *n*-caprylate to *n*-caproate (C) in three reactors across three main periods (and 10 periods). Period divisions are explained in the methods. Error bars indicate the standard error for the measurements. *Legend indicates periods (Periods 1D, 3B, and 3C) in which biomass samples were collected from reactors for shotgun metagenomic analysis. R1–3 are Reactors 1–3.

can reduce the H$^+$ produced by the pathway to H$_2$ (Fig. 1). The reactor tightness and material diffusiveness may influence the H$_2$ partial pressures because H$_2$, as the smallest molecule, may easily diffuse out of the system, while other gases would not. We built almost the entire reactor setup out of stainless steel to minimize H$_2$ diffusion through plastic tubing and connections. However, our results show that we could not prevent H$_2$ diffusion out of the system, which included a gas recirculation pump and some tubing lines not made of stainless steel. We note that we did not directly measure the reactor tightness and material diffusivity in this study.

The H$_2$ partial pressure can negatively affect chain elongation by reducing chain-elongating rates and changing the product spectra (21, 37). High H$_2$ partial pressures can lower the ethanol oxidation rate to acetate and prevent the Rnf complex from functioning properly within the RBOX pathway, resulting in lower ATP production through substrate-level phosphorylation and membrane-based phosphorylation, respectively (21). This would lower the growth rate, further slowing the development of an active microbiota. With the relatively high H$_2$ partial pressures for Reactor 3 during Period 1 compared to Reactors 1 and 2, a significant fraction of ethanol that we fed to Reactor 3 was not converted and left in the effluent, which resulted in a higher average effluent ethanol concentration for Reactor 3 compared to Reactors 1 and 2 (1.7 × 10$^2$ ± 9.7 mM vs 47 ± 3.9 mM and 29 ± 4.3 mM, respectively) (Fig. S2; Table S2). Because our reactors were continuously stirred systems, concentrations measured in the effluent were approximately equal to what the reactor microbiomes observed.

To test whether a lower H$_2$ partial pressure would improve *n*-caprylate production rates, we sparged N$_2$ gas into Reactor 3 to reduce the percentage of H$_2$ in the headspace (Table 1; Table S1). The sparging decreased the H$_2$ in the headspace from 31% ± 9.6% (by volume) during Period 1 to 20% ± 14% during Period 2 to 7.3% ± 4.6% during Period 3 (Table S1), resulting in increased volumetric *n*-caprylate production rates for Reactor 3 during Periods 2 and 3 (Fig. 2B). Into Reactor 2, we sparged N$_2$ and H$_2$ gas. As expected, when H$_2$ partial pressures increased during Periods 2 and 3 (Table S1), *n*-caprylate productivity decreased for Reactor 2 (Fig. 2B). However, we observed that the effect of H$_2$ on *n*-caprylate production was not uniform in all reactors. When the amount of H$_2$ in the headspace decreased due to N$_2$ sparging into Reactor 1, we observed decreased *n*-caprylate production rates during Periods 2 and 3 (Fig. 2B; Table S1). However, sparging with N$_2$ to remove H$_2$ may have also removed O$_2$, which could have an unknown effect.

Gas sparging itself was another introduced variable in the experiment that may have decreased biomass growth and *n*-caprylate production for Reactor 1 during Periods 2 and 3. We also noted differences in the acetate, *n*-butyrate, *n*-caproate, and *n*-caprylate concentrations in the effluent of our reactors (Table S2; Fig. S2A through C). Thus, our system was not predictive because we did not fully understand how the environmental conditions in the reactor affect the microbial pathways in the complex microbiota.

## Bacterial species abundance correlated with *n*-caprylate production rates

We analyzed the reactor microbiome via 16S rRNA gene sequencing and shotgun metagenomic sequencing. Overall, we observed similar trends in the dominance of certain bacterial species during high and low *n*-caprylate production periods in both data sets. We noticed some differences between the sequencing methods, which we attributed to differences in how the data were analyzed and how taxonomy was assigned (see Materials and Methods). The 16S rRNA gene sequencing data set was derived from approximately weekly biomass samples, which we collected from the reactors throughout the operating period. The shotgun metagenomics data set was smaller and was derived from nine biomass samples, which we collected from the three reactors at three-time points during the operating period (during Periods 1D, 3B, and 3C, as indicated in Fig. 3). The shotgun metagenomic data set resulted in 477,902,544 reads. We assembled 32 draft genomes from this data, 25 of which were high-quality (>90% completion, <5% contamination) (38), as detailed in Table 2.

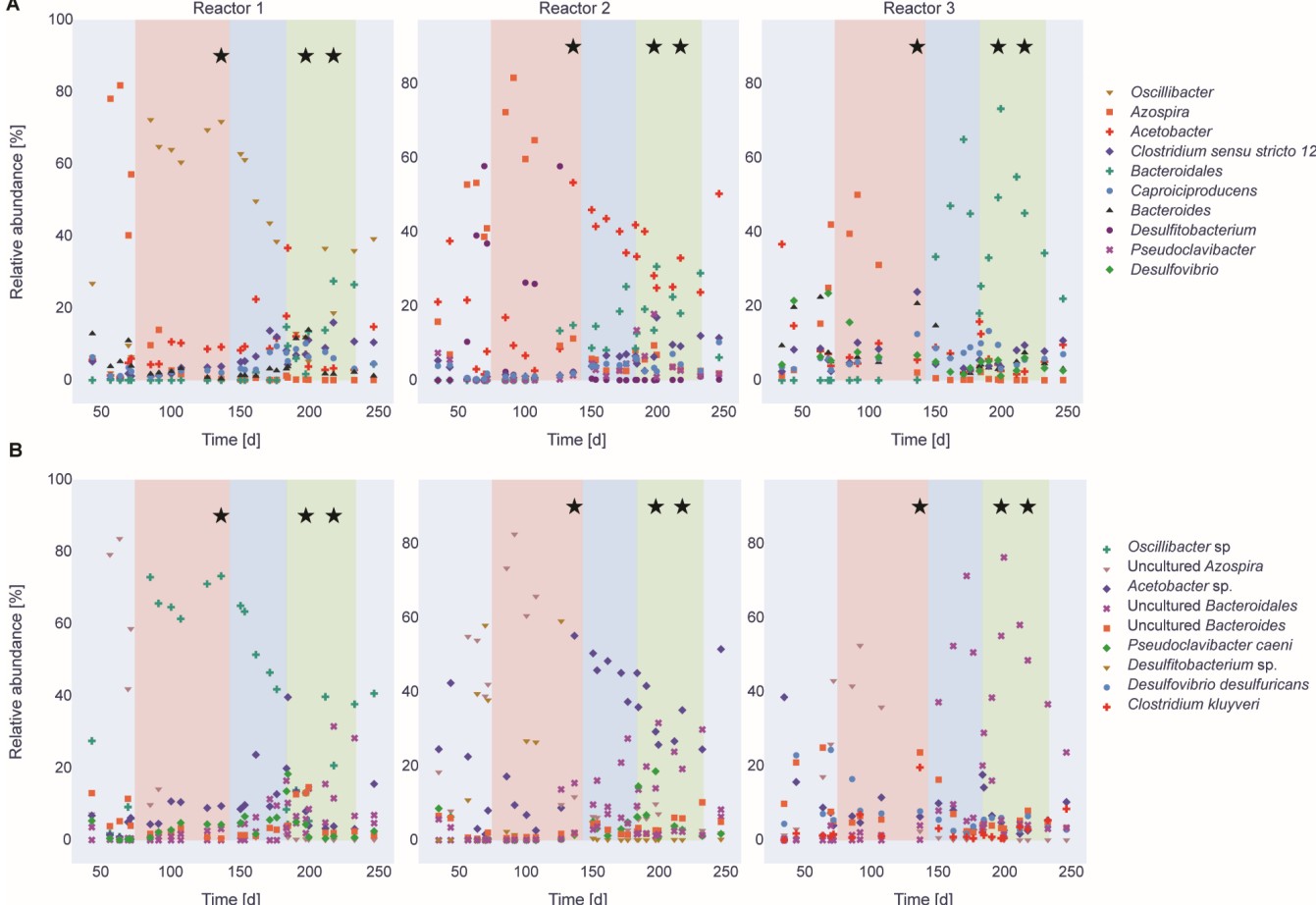

**FIG 3** Relative abundance of the top seven most dominant taxa of each reactor based on the Illumina 16S rRNA gene sequencing results on the genus level (A) and the species level (B) throughout the operating time. The first 75 days of the operating period were the startup period (light blue). The salmon, blue, and green shadings indicate Periods 1, 2, and 3, respectively; the stars indicate the metagenomic sampling time points.

For metagenome-assembled genomes (MAGs) with high contamination, we observed multiple instances of single-copy genes, likely from the same or closely related strains, indicating contamination.

For both the 16S rRNA gene sequencing data and the shotgun metagenomics data, certain bacterial species within the genus *Oscillibacter* dominated when *n*-caprylate production rates were higher for Reactor 1 during Period 1 and decreased in abundance during later periods when production rates decreased (Fig. 3B and 4). The unknown *Oscillibacter* sp. bacteria that was dominant in the 16S rRNA gene sequencing data had a 95.9% ID to an *O. valericigenes* Sjm18-20 strain (39). In the shotgun metagenomics analysis, *O. valericigenes* was one of the dominant bacteria in Reactor 1 during Period 1 (126,083 aligned reads, Fig. 4A). Based on the shotgun metagenomics analysis, *O. valericigenes* abundance positively correlated with *n*-caprylate production rates (Pearson correlation coefficient, r = 0.68, P = 0.0439). Several other *Oscillibacter* spp. were also positively correlated with *n*-caprylate production rates, *Oscillibacter* sp. CAG:155 (r = 0.71, P = 0.0321), *Oscillibacter ruminantium* (r = 0.65, P = 0.058), *Oscillibacter* sp. 1-3 (r = 0.72,

**TABLE 2** MAGs found in the reactors*a*

| MAG ID | GTDB | NCBI | Completeness | Contamination | Reactor/period |
|---|---|---|---|---|---|
| 1 | *Acetobacter* sp012517935 | *Acetobacter* sp. | 100 | 0.75 | R3/1D |
| 2 | *Methanobacterium_C* | *Methanobacterium* | 100 | 0.8 | R1/3C |
| 3 | JAAYAE01 | Acholeplasmataceae bacterium | 99.8 | 0.9 | R2/3B |
| 4 | *Intestinimonas* | *Intestinimonas* | 98.99 | 0 | R3/3C |
| 5 | *Desulfitobacterium* | *Desulfitobacterium* | 98.85 | 2.87 | R2/1D |
| 6 | *Pseudoclavibacter caeni* | *Pseudoclavibacter caeni* | 98.84 | 0.58 | R2/1D |
| 7 | *Cellulomonas* | *Cellulomonas* | 98.65 | 0.72 | R3/3C |
| 8 | JAAYUD01 sp012517855 | Bacteroidales bacterium | 98.49 | 0 | R3/3C |
| 9 | JAEWCM01 | Bacillota bacterium | 98.47 | 1.32 | R1/3B |
| 10 | *Bacteroides* | *Bacteroides* | 98.12 | 3.35 | R1/3B |
| 11 | *Bacteroides* sp900766195 | Uncultured *Bacteroides* sp. | 98.12 | 2.79 | R3/1D |
| 12 | *Clostridium* AM | *Clostridium drakei* | 98.09 | 4.16 | R2/3B |
| 13 | *Fimivivens* | Bacillota bacterium | 97.93 | 0.71 | R3/3C |
| 14 | *Prevotella* | *Prevotella* | 97.80 | 0.79 | R3/3C |
| 15 | JAAYSF01 | Veillonellaceae bacterium | 97.12 | 4.86 | R3/1D |
| 16 | *Desulfovibrio legallii* | *Desulfovibrio* | 97.04 | 0 | R3/3C |
| 17 | *Caproicibacterium* sp002411615 | Ruminococcaceae bacterium UBA5397 | 96.17 | 0.36 | R3/1D |
| 18 | *Latilactobacillus fuchuensis* | *Latilactobacillus fuchuensis* DSM 14340 = JCM 11249 | 95.55 | 0 | R1/3C |
| 19 | *Levilactobacillus brevis* | *Levilactobacillus brevis* ATCC 14869 = DSM 20054 | 94.84 | 0 | R1/3C |
| 20 | *Clostridium* B | *Clostridium kluyveri* DSM 555 | 94.72 | 0.69 | R3/1D |
| 21 | *Oscillibacter* | Clostridia bacterium | 94.15 | 2.85 | R1/3B |
| 22 | *Vescimonas* | Clostridiales bacterium | 93.62 | 2.01 | R2/1D |
| 23 | Oscillospiraceae UBA2922 | Clostridiales bacterium UBA2922 | 92.84 | 3.36 | R3/1D |
| 24 | *Pauljensenia* | Actinomyces | 92.04 | 3.79 | R3/1D |
| 25 | *Clostridium AV fermenticellae* | *Clostridium fermenticellae* | 90.01 | 15.84 | R1/1D |
| 26 | *Acetobacter fabarum* | *Acetobacter fabarum* | 89.38 | 1.74 | R3/1D |
| 27 | *Azospira* | *Azospira* | 89.34 | 0.36 | R2/1D |
| 28 | Clostridiaceae | Clostridiaceae | 85.88 | 0.84 | R3/3C |
| 29 | *Pygmaiobacter* | Bacillota bacterium | 83.09 | 1.38 | R1/1D |
| 30 | *Azospira inquinata* | *Azospira inquinata* | 76.76 | 0.84 | R2/3B |
| 31 | *Onthomonas* | *Clostridiales bacterium* | 75.88 | 2.35 | R2/3C |
| 32 | *Bulleidia* | *Solobacterium* | 75.45 | 2.22 | R1/3B |

*a*The table depicts a MAG identifier, the taxonomy of the bins assigned by GTDB-tk, and the corresponding NCBI name. Species names are displayed. If no species classification was found by GTDB-tk, the genus name is displayed. The completeness and contamination computed by CheckM, and the corresponding period in the reactor are also displayed. MAGs are ordered by their percentage of completeness.

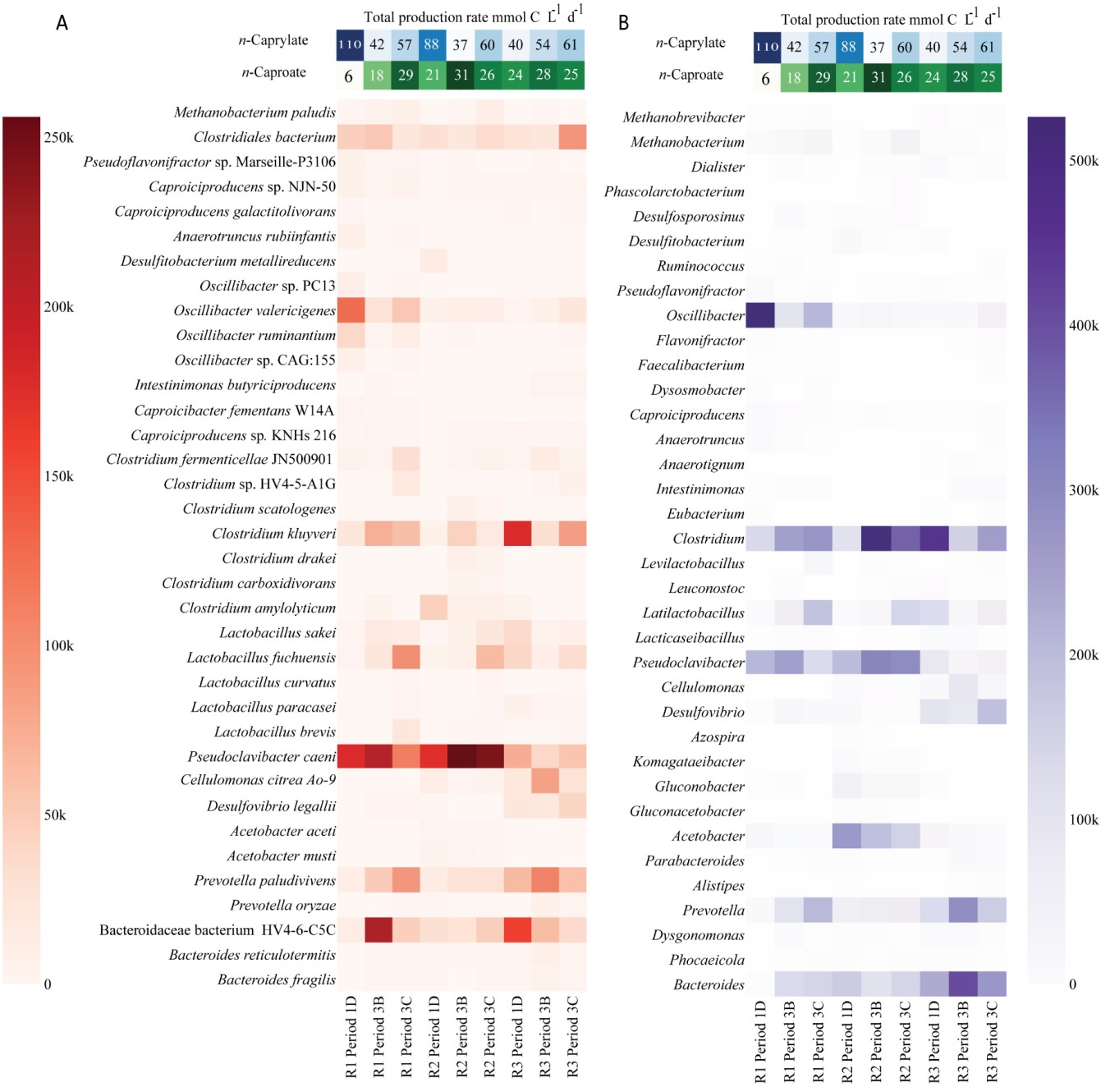

**FIG 4** The most abundant species (A) and genera (B) in the three reactors based on the shotgun metagenome analysis. After normalizing the read count for sample size, the heatmap shows the number of reads aligned to each taxon throughout different sampling points of the reactors. Only taxa with more than 12 k reads aligned are displayed. The names are ordered based on the NCBI taxonomy. The top of the plot shows the *n*-caprylate (blue) and *n*-caproate (green) volumetric production rates for Periods 1D, 3B, and 3C, respectively; color intensity is proportional to the production rates. R1–3 are Reactors 1–3.

$P = 0.0287$), *Oscillibacter* sp. NSJ-62 (r = 0.65, $P = 0.058$), and *Oscillibacter* sp. PC13 (r = 0.68, $P = 0.0439$) (Fig. 4). Based on the 16S rRNA gene sequencing data, an *Oscillibacter* sp. OTU, an uncultured *Oscillibacter* OTU, and an *O. valericigenes* OTU were positively correlated with *n*-caprylate production rates (r = 0.38, 0.27, and 0.15, and $P = 0.001, 0.021, 0.208$, respectively, Fig. 3).

Two aerobic bacteria, *P. caeni,* and an unknown *Acetobacter* sp., were present in some reactors at relatively high abundances. For Reactors 1 and 2, *P. caeni* was an abundant bacterium. Still, its abundance did not correlate to *n*-caprylate production rates (r = 0.01,

$P = 0.98$, Fig. 4). For Reactor 2, *Acetobacter* sp. bacteria were dominant during Periods 1 and 2 and declined in abundance during Period 3 when *n*-caprylate production rates decreased (Fig. 3 and 4). The presence of these bacteria shows that $O_2$ was introduced into the reactors due to an unknown location in the reactor setup.

Certain bacterial species dominated the reactors during periods of relatively low *n*-caprylate production but higher *n*-caproate production rates (Period 3B for Reactors 1 and 2 and Period 1D for Reactor 3; Fig. 2 to 4). Based on the shotgun metagenomics data, the abundance of *C. kluyveri* was negatively correlated to *n*-caprylate production rates. However, the correlation was not significant (Fig. 4, r = −0.49, P = 0.18). No correlation was observed between *C. kluyveri* relative abundance and production rates in the 16S rRNA gene sequencing data (Fig. 3B). For Reactor 1 during Period 3B, Bacteroidaceae bacterium HV4-6-C5C (217,314 aligned reads) and *P. caeni* (210,785 aligned reads) dominated the reactor. For Reactor 2 during Period 3B, *P. caeni* (255,949 aligned reads) and, to a lesser extent, *C. kluyveri* (43,228 aligned reads) dominated the reactor. For Reactor 3 during Period 1D, *C. kluyveri* (180,084 reads) and Bacteroidaceae bacterium HV4-6-C5C (157,401 reads) dominated the reactor (Fig. 4). Across all reactors, the abundance of Bacteroidaceae bacterium HV4-6-C5C was negatively correlated to *n*-caprylate production rates, though the correlation was not significant (r = −0.54, P = 0.133 Fig. 4). Based on the 16S rRNA gene sequencing data, an *Azospira* sp. was dominant in all the reactors prior to and at the start of Period 1 and was positively correlated to *n*-caproate production rates (r = 0.41, P = 0.00034 Fig. 3).

## Bacteria with the RBOX pathway

We investigated which metagenomes in our reactors had a complete or nearly complete RBOX pathway (Fig. 5). Specifically, we looked for nine enzymes involved in the RBOX pathway in our metagenomics and proteomics data: acetate CoA-transferase (CoAT), 3-hydroxy-acyl-CoA dehydrogenase (HAD), enoyl-CoA dehydratase (ECH), acyl-CoA dehydrogenase (ACD), electron-transfer-flavoprotein subunit A/B (EtfA/B), acetyl-coenzyme A acetyltransferase (ACAT), thioesterase (TE), and Rnf respiratory complex (RNF) (Fig. 1).

All bacteria with the complete RBOX pathway in their metagenome and proteome were in the class *Clostridia*, except for *Azospira inquinata* and JAAYAEO1 (NCBI classification: Acholeplasmataceae bacterium) (Fig. 5). In the 16S rRNA gene sequencing data, an OTU classified as the *Clostridia* class member *O. valericigenes* dominated Reactor 1 during periods of high *n*-caprylate production (Fig. 3). A MAG classified as *Oscillibacter* contained all the RBOX enzymes in its metagenome and proteome (Fig. 5). The other bacteria in the class *Clostridia*, which had the complete RBOX pathway, were not dominant: *Clostridium* AV *fermenticellae*, *Vescimonas*, *Desulfitobacterium*, *Clostridium* AM, *Caproicibacterium* sp002411615, and *Fimivivens* (Fig. 5). In a prior study, which utilized inoculum from this study's reactors, a *Caproiciproducens* strain (7D4C2) was isolated from reactor biomass and shown to produce *n*-caproate (40). In our reactors, the metagenome of *Clostridium* B (NCBI: *C. kluyveri* DSM 555 Fig. 5; Table 2) contained and expressed the majority of RBOX enzymes except for acetyl CoA-transferase (CoAT), which was not found in the proteome, and electron-transfer-flavoprotein subunit B (EtfB), which was not found in the proteome or metagenome (Fig. 5). It is important to note that the absence of a protein in our comparative proteomics does not mean that the protein was not expressed as some proteins might not have been detected.

## Bacterial microcompartments present in reactor microbiomes

The metagenomic analysis revealed the presence of specific bacterial microcompartments. These specialized compartments encapsulate metabolic pathways, enhancing metabolic efficiency and specificity. It is known that the chain-elongator *C. kluyveri* and other Clostridia have microcompartments to protect their intracellular milieu against unstable or toxic chemical intermediates (41). Notably, an ethanolamine utilizing microcompartment (EUT2B), two propanediol utilizing microcompartments (PDU1D and

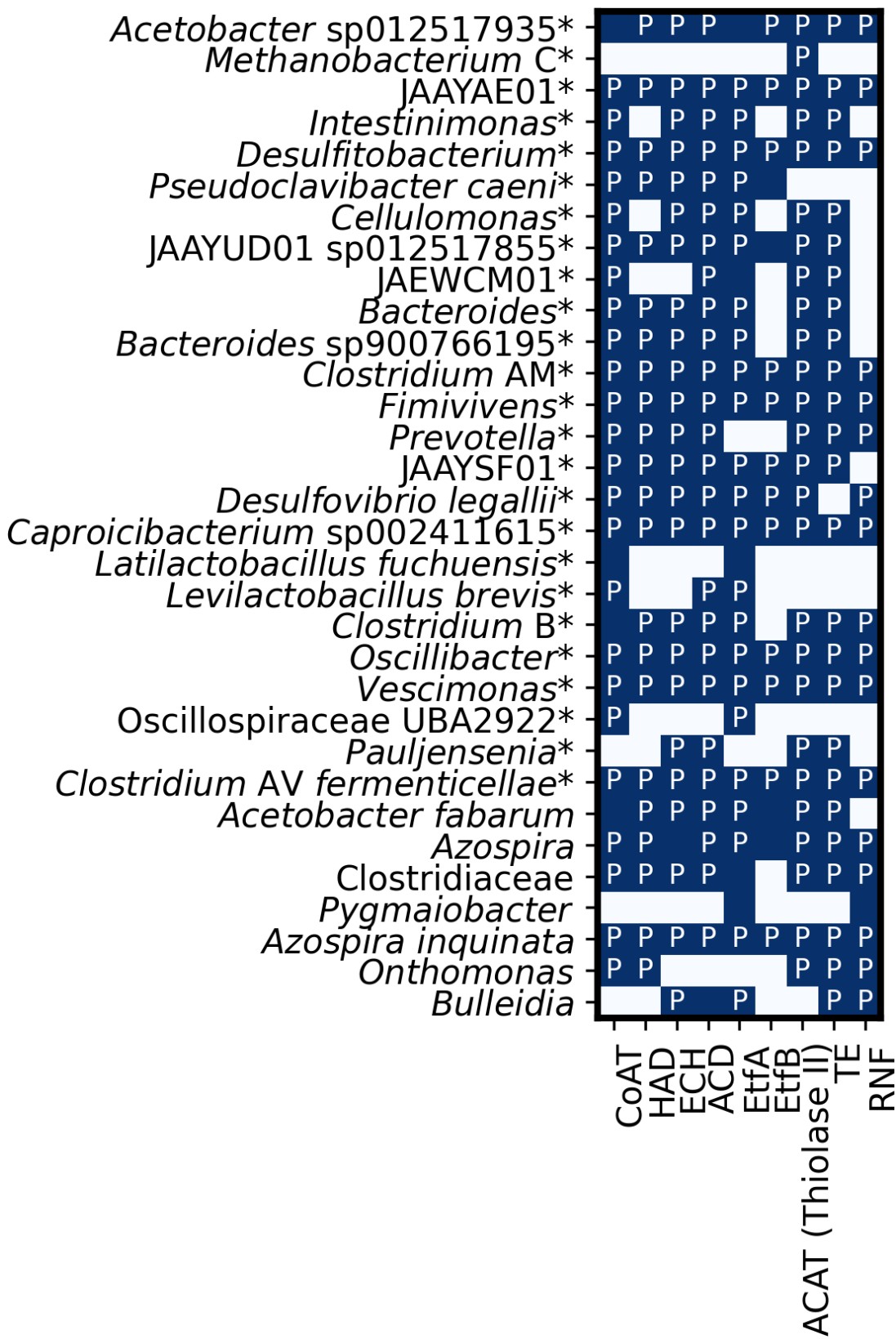

**FIG 5** Absence or presence of nine enzymes involved in the RBOX pathway (acronyms described previously) in reactor *de-novo* assembled metagenomes and proteomes as monitored by shotgun metagenomics analysis and proteomics. MAG taxonomy was assigned using gtdb-tk (r220). Enzyme acronyms were described in Fig. 1. A blue box denotes the presence in the metagenome, the letter P the presence in the metaproteome, and a white box without a P the absence in both the metagenome and metaproteome. MAGs identified to species level are depicted. A * indicates high-quality MAGs >90% complete and <5% contaminated (determined with CheckM).

PDU1C), and a glycyl radical enzyme containing microcompartment (GRM1A) were the dominant microcompartments observed (Fig. S3). These microcompartments are metabolosomes that are usually expressed only when their substrate is present (42). The ethanol-utilizing microcompartment (ETU) metabolizes ethanolamine, which is a product of the breakdown of phosphatidylethanolamine, to ethanol, acetyl-CoA, and acetyl-phosphate and protects the rest of the bacterial cell from the intermediate acetaldehyde (43). We also observed the presence of ETU in all time points studied (Fig. S3). This bacterial microcompartment has only been reported in the bacterium *C. kluyveri* (44, 45). We observed the ETU microcompartment as expected in *C. kluyveri*. We also observed the ETU microcompartment in other bacteria classified to the level *Clostridium* spp. or Clostridiaceae (Fig. S3), though we note that these could be *C. kluyveri* species that cannot be classified to the species level. We also observed the ETU microcompartment in proteins that had no hit in the taxonomic database (NAs in Fig. S3). *O. valericigenes* Sjm18-20, which dominated Reactor 1 during periods of high *n*-caprylate production, was not found to have ETU microcompartments, though, it did contain EUT2B, PDU1D, and GRM1A, GRM3A, and GRM3C microcompartments.

## DISCUSSION

For our open-culture reactors, different microbial communities were correlated with periods of high *n*-caproate or *n*-caprylate production (Fig. 4). The known chain elongator *C. kluyveri* and a primary fermenter Bacteroidaceae bacterium HV4-6-C5C had higher relative abundances during periods of high *n*-caproate production and decreased in abundance during periods of high *n*-caprylate production (Fig. 3 and 4). *C. kluyveri* produces *n*-caproate from ethanol and short-chain carboxylates (acetate or *n*-butyrate), but there is limited evidence of its ability to produce *n*-caprylate (10). In particular, the ethanol-utilizing microcompartment, ETU, associated with *C. kluyveri*, underscores its potential role in the efficient conversion of ethanol to acetaldehyde, which is a pivotal step in the production of *n*-caproate and *n*-caprylate. Different *Oscillibacter* species, which include *O. valericigenes* (r = 0.68, P = 0.0439), were positively correlated to periods of high *n*-caprylate output in the reactors (Fig. 3 and 4). Indeed, *O. valericigenes* included microcompartments to possibly protect themselves during chain elongation. This finding is consistent with prior studies for which members of the Ruminococcaceae family (to which *Oscillibacter* belongs) were isolated from reactors producing *n*-caproate from lactate (9, 29) and Illumina 16S rRNA gene sequencing studies for which Ruminococcaceae members were associated with MCC production in reactors (4, 7, 11).

The unplanned presence of $O_2$ in our reactors created a niche for aerobic bacteria, such as *P. caeni* and *Acetobacter* species, to survive and become abundant in the reactors (Fig. 4). The abundance of these aerobic bacteria was not correlated to *n*-caprylate production rates (Fig. 3 and 4). As a result of our inability to build a reactor system that prevented $O_2$ inclusion, a major caveat existed in our quest to study different $H_2$ partial pressures on the RBOX. Using gas sparging to remove or add $H_2$ also removed $O_2$, which was a sensitive parameter. Even though we could not satisfy our experimental design with the independent parameter $H_2$, this study provides information on which to base future research, as discussed below. Aerobic or facultative anaerobic microbes must have quickly consumed the $O_2$ in our reactors because strict anaerobic microbes, such as methanogens and other obligate anaerobes, were also present in our continuously stirred reactor systems (Fig. 4). Prior studies observed aerobes, such as *Acetobacter* (3, 11), and facultative anaerobes, such as *Lactobacillus* (11, 46), in chain elongation reactors. Previous studies from our lab had not found *P. caeni* in similar chain-elongating reactors (3, 4), though the aerobe *Acetobacter* was observed (3). *P. caeni* was isolated from sewage sludge in 2012 (47), but the *P. caeni* assembly was only added to the NCBI nr database in 2019 (ASM883112v1). *P. caeni* could have been present in previous reactor studies but not detected due to its absence from existing databases. A previous study from 2016 found a phylotype that matches *P. caeni* in batch experiments utilizing biomass from

a chain-elongating reactor fed a variety of substrates and found its occurrence did not correspond to chain elongation activity (48).

From our metagenomic and metaproteomic analyses, we conclude that the RBOX pathway was active in our reactors (Fig. 5). Some abundant bacteria did not have the complete RBOX pathway (Fig. 5), which may indicate: (i) that our methods did not always identify all genes or proteins; or (ii) that chain elongators live in syntropy with each other to produce the medium-chain carboxylic acids. We observed that RBOX was affected by the partial pressures of $H_2$ in the headspace of the reactors, which follows the current understanding of chain elongation. The presence and distribution of specific bacterial microcompartments in the dominant bacteria (Fig. S3) could influence this observed metabolic pathway, reflecting the potential versatility introduced by these microcompartments. We should note that we only have evidence that genes for the bacterial microcompartments were present in the genomes, not that they were expressed. As expected, *C. kluyveri* (*Clostridium* B) had most of the RBOX pathway in the metagenome and proteome (Fig. 5). Some *Clostridia* class members had the complete RBOX pathway in their metagenome and proteome, including an *Oscillibacter* species (Fig. 5). We also note that some bacteria found in our study, specifically *A. inquinata* and JAAYAE01 (Acholeplasmataceae bacterium) had the complete RBOX pathway (Fig. 5), but are not known chain elongators. Previous researchers have also noted the presence of the RBOX pathway in bacteria that are not known chain elongators (40).

Our study provides insight into the bacteria producing *n*-caproate and *n*-caprylate from ethanol and acetate via RBOX. We identified potential candidates for *n*-caprylate production in our reactors. Future studies should try to isolate and sequence *n*-caprylate-producing bacteria. In addition, future studies should also investigate whether a relative lack of diversity in *n*-caprylate-producing reactors affects the stability of these systems. The potential influence of bacterial microcompartments on this metabolic pathway, as observed in our shotgun metagenomic analysis, underscores the need to consider their role in future studies. Future research should also investigate the role of microaerobic conditions in these reactors because we observed that $O_2$ is a sensitive parameter, but we do not know why.

## MATERIALS AND METHODS

### Continuously fed reactor system

We designed, self-built, and operated three grade-316 stainless steel reactors with a 5.5 L total volume (5 L working volume and 0.5 L headspace volume) (parts utilized in the building system are detailed in Table S3). We maintained the reactor pH at ~5.5 (via periodic additions of 0.5 M HCl) and the temperature at 30°C ± 1.0°C. The reactors were continuously mixed via a peristaltic pump (Cole Parmer, Part No. 7520-10), which recirculated the reactor broth at a rate of ~40 mL min$^{-1}$ by removing broth from the top of the reactor liquid level and returning it to the reactor base (internal recycle line; Fig. S1). We continuously fed the reactors with a modified-based media that was previously described (4, 49) and supplemented with ethanol and acetate. After a 75-day startup period, we mixed broth from all reactors to ensure similar microbiota in each reactor before a restart. We operated the reactors as replicates in which we kept organic loading rates and hydraulic retention time (HRT) at $1.5 \times 10^2$ ± 4.6 mM C L$^{-1}$ d$^{-1}$ and 8.5 ± 0.2 days, respectively, for a period of 68 days (Period 1 of study—Days 75 to 142; see Table 1). This organic loading rate was lower than in prior studies in our lab with ethanol and acetate-fed reactors (3, 4). Still, the different reactor designs should be noted (i.e., continuously mixed reactors in this study vs upflow anaerobic filters in the prior studies). The solids retention time was not measured in our reactors. During Periods 1 to 3 of the study, the molar ratio of ethanol to acetate was maintained at 10:1 in the substrate, and the ethanol concentration was ~600 mM.

Reactors were inoculated with 10% by volume (~500 mL) of reactor broth from a reactor that was fed semi-continuously (~once every 2 days) with ethanol-rich yeast

fermentation beer and operated as an anaerobic sequencing batch reactor for an operating period of approximately 5 years prior to the time we collected the inoculum (7, 16). For in-line product extraction, we used a setup such as the one previously described by Agler et al. (7). Detailed information on the reactor setup can be found in the Supporting Information. During Periods 2 and 3, gases ($N_2$ and $H_2$) were sparged into the bottom port of the reactors (Table 1).

## Experimental periods for reactors

The primary study periods were Periods 1 (Days 75 to 142), Period 2 (Days 143 to 184), and Period 3 (Days 185 to 234), and were divided into periods 1A—Days 75 to 91, 1B—Days 92 to 113, 1C—Days 114 to 125, 1D—Days 126 to 142, 2A—Days 143 to 151, 2B—Days 152 to 162, 2C—Days 163 to 184, 3A—Days 185 to 194, 3B—Days 195 to 206, and 3C—Days 207 to 222 (Table 1). Prior to Period 1, there was a 75-day startup period for the reactors in which the organic loading rate was incrementally increased to the target loading rate of ~$1.4 \times 10^2$ mM C $L^{-1}$ $d^{-1}$ at an HRT of ~9 days. At the start of Period 1 (Day 75), biomass from all three reactors was combined, mixed, and redistributed. During Period 1, operating conditions (i.e., temperature, pH, product extraction) were kept the same. During Period 2, gas sparging of $N_2$ gas was tested out (i.e., gas sparging was off and on irregularly between Days 143 to 184) (Table 1). At the start of Period 3 (Day 185), biomass was again mixed and redistributed. During Period 3, we sparged Reactor 1 and Reactor 3 continuously with $N_2$ gas, while we sparged Reactor 2 continuously with a mixture of $H_2$ and $N_2$ gas (Table 1). Although we did not measure the gas flow rate that we sparged into the reactors during Period 3, sparging rates are assumed to be equal to the measured exit gas flow rates reported due to low gas production rates that we observed during Period 1 without sparging (Table 1). Throughout Periods 1 to 3, we aimed for similar organic loading rates to all reactors. Relatively small differences in organic loading rates (Table 1) can be attributed to minor differences in the influent flow rate and prepared influent composition supplied to the three reactors.

## Liquid and gas analysis

We collected liquid samples from reactor broth and alkaline extraction solution to measure carboxylate and ethanol concentrations. The 2 mL samples of reactor broth were collected from a port in the reactor system recycle line. In contrast, we collected the alkaline extraction solution samples from a ~3 L well-mixed glass reservoir from which the extraction solution was re-circulated. Samples were stored frozen at −20°C prior to analysis. Gas chromatography systems were used to determine carboxylate and ethanol concentrations, as has been described by Usack et al. (50). We collected gas samples from the gas exit lines of the reactors. $CO_2$, $CH_4$, and $H_2$ concentrations (>0.2% by volume) were measured using a gas chromatography system, which has been described previously (50). A reduction gas detector was used to measure $H_2$ gas concentrations <0.2%, which has been described by Kucek et al. (4).

## Calculations and statistical analysis of operating data

We calculated the carboxylate production rates as average values for each operating period. We summed the average effluent production rates per liter of the reactor (mmol C $L^{-1}$ $d^{-1}$) and the average transfer rates via product extraction (mmol C $L^{-1}$ $d^{-1}$) to yield total production rates per liter of the reactor (mmol C $L^{-1}$ $d^{-1}$). We calculated the average effluent production rates by dividing the average carboxylate concentration per period by the average HRT. We calculated the average HRT per period based on the average effluent flow rate per period, which was determined gravimetrically. We calculated the average transfer rates by plotting the increasing concentrations of individual carboxylates in alkaline extraction solution vs time. We used least squares methods to determine the slope and the sample standard deviation (LINEST function, Microsoft Excel). We divided the slope by the reactor working volume (5 L) to obtain an average transfer rate

per period. RStudio v.1.0.136 (51) was used to run data analysis in R. Concentrations, rates, ratios, and efficiencies are reported as mean value ± standard error in the paper unless noted otherwise.

## 16S rRNA gene sequencing analysis

We took close-to-weekly biomass samples for Illumina 16S rRNA gene sequencing analysis from the internal recycle line of the reactors. Approximately 10 mL of reactor broth was collected with a 60 mL plastic syringe and distributed into 2 mL Eppendorf tubes. We centrifuged the tubes at 16,873 × $g$ for 4 min and discarded the supernatant. Finally, we stored the pelleted biomass samples at −80°C.

According to the manufacturer's protocol, we extracted genomic DNA using the PowerSoil-htp 96 Well Soil DNA Isolation kit (MO BIO Laboratories Inc., Carlsbad, CA, USA). PCR amplification with 515-forward and 806-reverse Golay barcoded primers targeting the V4 region of the 16S rRNA gene of the extracted DNA was described previously (52) with the following exceptions: Mag-Bind RxnPure Plus magnetic beads solution (Omega Biotek, Norcross, GA, USA) was used instead of Mag-Bind E-Z Pure, and 50 ng DNA per sample was pooled instead of 100 ng. Duplicate PCR reactions of each DNA extract were performed and pooled prior to sequencing. Samples were sent for paired-end sequencing (2 × 250 bp) on the Illumina MiSeq platform (Illumina, San Diego, CA, USA) at the Cornell University Biotechnology Resource Center (Ithaca, NY, USA). We analyzed the resulting 16S rRNA gene sequencing reads using QIIME 2 2017.3 (53) and the Silva database release 138.1. Finally, we investigated the correlation of the relative abundance for OTUs with $n$-caproate and $n$-caprylate production rates using the scipy-stats package pearsonr (54).

## Shotgun metagenomic analysis

We collected biomass samples for shotgun metagenomic analysis approximately weekly from internal liquid-recycle lines of the reactors, which were utilized to mix the reactor liquid. Samples were centrifuged, supernatant was discarded, and biomass was stored at −80°C. Genomic DNA was extracted using the PowerSoil DNA Isolation kit (MO BIO Laboratories Inc.). We used a modified protocol, which has been described by Kucek et al. (4). After quantifying the extracted DNA, we selected nine samples for shotgun metagenomics sequencing (three samples for each reactor during Periods 1C, 3B, and 3C). For Period 1C, we selected one sample from Reactors 1 and 2 on Day 137 and a pooled sample from Reactor 3 from Days 137, 151, 154, and 162. For Period 3B, we selected a pooled sample from each reactor on Days 198 and 200. For Period 3C, we selected one sample from each reactor on Day 218. Pooled samples were utilized if the concentration of the genomic DNA extracted was low on a single day. The nine selected DNA samples were barcoded and sequenced on two lanes (100 bp per read; single-direction reads) using the Illumina HiSeq platform at the JP Sulzberger Genome Center at Columbia University (New York, NY, USA). We merged the replicates of samples.

Shotgun metagenomics read quality was checked using FastQC (55) after trimming with Trimmomatic (56). We performed quality control using FastQC (55) version 0.11.9 on merged reads. The sequence quality scores and histograms passed standard test criteria for all samples (lower quartile for every base above 10 and median above 25). To trim low-quality regions and remove low-quality reads, Trimmomatic (56) version 0.39. was applied on all samples providing the parameters -phred33; LEADING:3; TRAILING:3 as well as SLIDINGWINDOW::4:15 and MINLEN:36. Trimmed reads were aligned to NCBI-nr database (Feb 2021) using DIAMOND (57) version 2.0.7.in blastx mode. The following parameters were used: --outfmt 100 -c1 -b12 -p 32 --top 10 -e 0.001. Resulting alignments were meganized for further analysis using daa-meganizer, which is a tool that is included in MEGAN6 (58). DIAMOND output files were loaded into MEGAN6, were normalized by sample size, and read counts were extracted for each MAG. Heatmaps were created using a Python script, only displaying MAGs with more than 12 k aligned

reads (Fig. 4). The correlation of read counts with *n*-caproate and *n*-caprylate production rates was investigated using the Pearson correlation coefficient (54).

## *De-novo* assembly

We performed *de-novo* assembly for each set of quality filtered reads using MEGAHIT (59) version 1.2.9 with preset meta-large for large and complex metagenomes. This resulted in 757,643 contigs with a mean length of 1,129 bp and a mean N50 of 2,620 bp. Assembled contigs were binned using MetaBAT 2 with default parameters (60). The resulting bins from all samples were then dereplicated using dRep2 (61). The best dereplicated bins were then checked for quality using CheckM (59) and taxonomy was assigned using GTDB-tk (release 220) (62). A summary of the assembled MAGs can be found in Table 2. We annotated the contigs using Bakta (63). We created a hidden Markov model (HMM) to search for the absence and presence of genes involved in the RBOX pathway. We based the included genes on the models used by Scarborough et al. (25). Pre-trained HMMs were downloaded from PFAM (64), and all annotated proteins from each sample were searched with these models using HMMER3 (65). For the RBOX pathway, PFAM only has 3-hydroxyacyl-CoA dehydrogenase (HAD) and not 3-hydroxy-butyrl-CoA dehydrogenase (HBD), so HBD was omitted. A custom Python script was used to plot the presence or absence of each gene in the annotated proteins. The annotated proteins were also searched for biological microcompartment (BMC) proteins using the BMC Caller tool (66) and plotted with Python (Fig. S3).

The samples from Periods 1D and 3C for Reactor 1 failed the per-base-sequence-content test, while samples from: (i) Periods 1D and 3C for Reactor 1; (ii) Period 1C for Reactor 2; and (iii) Period 3B for Reactor 3 all failed the per-sequence-GC-content test. Sequence duplication level was high in all samples except for the sample from Period 3C from Reactor 2. This problem was introduced when merging two replicates for each sample and is an artifact. Overall, the per-sequence quality scores were sufficient.

## Metaproteomic analysis

For metaproteomic sampling, approximately 200 mL of reactor broth was collected from the internal recycle lines of the reactors and distributed into four 50 mL centrifuge tubes. After centrifugation for 10 min at 8,000 $\times$ $g$ (at 4°C), the supernatant was discarded. Pellets were resuspended in a tris buffer solution and redistributed to 2 mL Eppendorf tubes. We spun the tubes for 4 min at 16,873 $\times$ $g$ and discarded the supernatant. Next, we stored the pelleted samples at $-20$°C.

Protein samples were extracted from reactor cell pellets (~100 µL bulk volume) using a gel-free, precipitation-free method to avoid loss of hydrophobic proteins. Cell pellets were suspended in 500 µL 50 mM Tris buffer (pH 8.0) and flash frozen 3$\times$ with liquid $N_2$ as an initial lysis step. 0.1% SDS, 10 mM NaCl, 0.02 M TCEP, and 2 M urea were added to lyse the sample by ultrasonication on the ice at 60% amplitude for 5 min total pulse time, vortexed, and centrifuged 10 min at 12,000 $\times$ $g$. Half of the supernatant (~250 µL) was removed and saved. To attempt to desorb more hydrophobic proteins from the pellet, 250 µL of acetonitrile was added to the cell pellet and the remaining supernatant. This was then vortexed and pelleted, while supernatant from this step was removed and re-combined with the first 250 µL of supernatant. The volume of the combined supernatant was decreased to approximately 400 µL via speed vac. We discarded the insoluble pellet. Total protein estimates were measured using the Bradford assay. Protein samples were reduced with an additional 0.05 M TCEP in 0.1 M ammonium bicarbonate at 35°C for 1 h, alkylated with 40 mM iodoacetamide at room temperature for 30 min, and digested with Pierce Trypsin Protease MS-Grade at an estimated 1:20 trypsin:protein mass ratio for 12 h at 35°C with 1 mM $CaCl_2$. Sample protein precipitation was avoided during digestion by diluting trypsin protease in 0.1 M ammonium bicarbonate buffer containing 0.02% SDS and 10% acetonitrile before combining with the protein sample. To quench digestion, samples were acidified to a pH of 3.5 with formic acid, acetonitrile was removed via speed-vac, acidified again to pH 3.5, and stored at $-20$°C. Tryptic

peptides were purified using 1 mL Supelclean ENVI-18 SPE tubes and dissolved in 0.1% TFA/0.5% acetonitrile for analysis by liquid chromatography-mass spectrometry (LC-MS).

LC-MS was performed using a Thermo Fisher UltiMate 3000 LC and LTQ-XL mass spectrometer with a standard ESI source. Microflow chromatography was performed on an Acclaim PepMap 100 column (1 mm × 15 cm; 3 um) at 40 µL/min using a 125 min gradient from 100% water (1% formic acid) to 40% acetonitrile. We operated the LTQ-XL in a 3× double play mode with a 10 s dynamic exclusion time and CID activation. The resulting peptides were compared to a decoy search. Peptides were thrown out based on a probabilistic filter. Proteins were kept if they had at least two unique peptides IDed with high confidence. The resulting 341 protein sequences were aligned against NCBI-nr (Feb 2021) using DIAMOND blastp version 2.0.7 for taxonomic assignment. To check for proteins involved in RBOX, we searched for these proteins using the previously described HMM models.

## ACKNOWLEDGMENTS

The authors would like to acknowledge Chase Brett and Doug Caveney for their help with constructing the reactor and Alex Marzelli and Dr. Jiajie Xu for assistance with reactor maintenance (all from Cornell University). We acknowledge funding from the U.S. EPA STAR grant fellowship, the U.S. Army Research Laboratory, and the U.S. Army Research Office under contract/grant number W911NF-12-1-0555. We also acknowledge funding from the Alexander von Humboldt Foundation in the framework of the Alexander von Humboldt Professorship to L.T.A., the Novo Nordisk Foundation $CO_2$ Research Center with grant number NNF21SA0072700 to L.T.A., the Deutsche Forschungsgemeinschaft under Germany's Excellence Strategy (EXC 2124–390838134) to L.T.A. and D.H. A special thanks go to the Reinhard Frank Stiftung to support the exchanges between the University of Maryland and the University of Tübingen.

## AUTHOR AFFILIATIONS

[1]Department of Biological and Environmental Engineering, Cornell University, Riley-Robb Hall, Ithaca, New York, USA

[2]Office of Undergraduate Research, University of Maryland, College Park, Maryland, USA

[3]Institute for Bioinformatics and Medical Informatics, University of Tübingen, Tübingen, Germany

[4]Department of Geosciences, University of Tübingen, Tübingen, Germany

[5]Chemistry Department, SUNY-Cortland, Bowers Hall, Cortland, New York, USA

[6]AG Angenent, Max Planck Institute for Biology Tübingen, Tübingen, Germany

[7]Department of Biological and Chemical Engineering, Aarhus University, Aarhus, Denmark

[8]The Novo Nordisk Foundation CO2 Research Center (CORC), Aarhus University, Aarhus, Denmark

## AUTHOR ORCIDs

Byoung Seung Jeon  http://orcid.org/0000-0002-8769-1603
Largus T. Angenent  http://orcid.org/0000-0003-0180-1865

## FUNDING

| Funder | Grant(s) | Author(s) |
| --- | --- | --- |
| U.S. Environmental Protection Agency (EPA) | STAR Fellowship | Catherine M. Spirito |
| Alexander von Humboldt-Stiftung (AvH) | Alexander von Humboldt Professorship | Largus T. Angenent |

| Funder | Grant(s) | Author(s) |
|---|---|---|
| Deutsche Forschungsgemeinschaft (DFG) | EXC 2124 - 390838134 | Daniel H. Huson |
| | | Largus T. Angenent |
| Novo Nordisk Fonden (NNF) | NNF21SA0072700 | Largus T. Angenent |
| DOD | USA | AFC | CCDC | Army Research Office (ARO) | W911NF-12-1-0555 | Largus T. Angenent |
| Reinhard Frank Stiftung | Collaboration between the University of Maryland and the University of Tübingen | Catherine M. Spirito |
| | | Largus T. Angenent |

## DATA AVAILABILITY

16S rRNA gene sequences are available at EBI (https://www.ebi.ac.uk/) under accession number ERP024135. Sequences and study metadata are publicly available in QIITA (https://qiita.ucsd.edu/) under study number 11227. Shotgun metagenomics data is available at SRA (https://www.ncbi.nlm.nih.gov/sra) under the accession PRJNA824684. Metagenomics and metaproteomics data analysis code is available on GitHub (https://github.com/lucass122/caprylate_reactor_paper).

## ADDITIONAL FILES

The following material is available online.

### Supplemental Material

**Data Set S1 (mSystems00416-24-s0001.csv).** RBOX gene and protein counts for all high-quality MAGs.
**Data Set S2 (mSystems00416-24-s0002.csv).** Bacterial microcompartments (BMCs) that were identified in reactor metagenomics samples using BMC caller.
**Supporting Information (mSystems00416-24-s0003.pdf).** Supplemental figures and tables plus additional methodology.

### Open Peer Review

**PEER REVIEW HISTORY (review-history.pdf).** An accounting of the reviewer comments and feedback.

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
