## [Reviewer comments · mSystems]

Variability in *n*-caprylate and *n*-caproate producing microbiomes in reactors with in-line product extraction.

Catherine Spirito, Timo Lucas, Sascha Patz, Byoung Seung Jeon, Jeffrey Werner, Lauren Trondsen, Juan Guzman, Daniel Huson, and Largus Angenent

Corresponding Author(s): Largus Angenent, Eberhard Karls Universitat Tubingen

Review Timeline:

Submission Date:	March 22, 2024
Editorial Decision:	April 14, 2024
Revision Received:	June 15, 2024
Accepted:	June 18, 2024

Editor: Yu-Liang Yang

Reviewer(s): Disclosure of reviewer identity is with reference to reviewer comments included in decision letter(s). The following individuals involved in review of your submission have agreed to reveal their identity: Matthew Scarborough (Reviewer #1)

Transaction Report:

DOI: <https://doi.org/10.1128/msystems.00416-24>

Re: mSystems00416-24 (The gases H₂ and O₂ in open-culture reactors influence the performance and microbiota of chain elongation into n-caproate and n-caprylate.)

Dear Prof. Largus T Angenent:

Revision Guidelines

Sincerely,
Yu-Liang Yang
Editor
mSystems

Reviewer #1 (Comments for the Author):

In this manuscript by Spirito, et al., the authors use metagenomics and proteomics to assess the impact of oxygen and hydrogen on a "chain elongation" bioreactor. Chain elongation has emerged as an area of significant research interest to convert organic wastes into beneficial products, however strategies to control microbiome behavior - especially controlling the fermentation end products -- remain elusive. Open-culture chain elongation is also a fantastic model system for understanding conversion processes in anaerobic microbiomes. Therefore, this manuscript should be of interest to readers of mSystems.

Within the specific field of chain elongation, the authors seek to answer an important question on the actual impacts of hydrogen and oxygen gas on the chain elongation process. The first is important because hydrogen is known to be create thermodynamic bottlenecks in anaerobic processes, but it has been suggested that hydrogen accumulation can be a positive in chain elongation by inhibiting competing biochemical transformations. Further, hydrogen has been proposed as an electron donor to improve chain elongation, but the evidence on the role of hydrogen is mixed thus far. Less interest has been paid to oxygen, but I have personally observed that microbial communities flanking chain elongators often include oxygen-tolerant species. Given that these are open-culture systems with complex feedstocks that are not pre-reduced, oxygen likely impacts the microbial communities. Therefore, I think that is an important topic for the field of chain elongation.

The paper is well-written, the analyses are well described, and the conclusions are justified. My comments relate mostly to the metagenomic approaches employed which can be improved. I provide point-by-point comments below.

Introduction, Paragraph 2: Please check this first sentence. There is not any substrate-level phosphorylation occurring by acetaldehyde dehydrogenase or alcohol dehydrogenase. Also, ATP may be produced through substrate level phosphorylation with acetate kinase. ATP is not needed to run RBOX-actually, the model you show later on (Figure 1) is great, so if the description matches these, it'll be perfect.

Introduction, Paragraph 2: Your interpretation of the FAB pathway is perfectly correct and I wish it could be bolded or highlighted and shared widely.

Results, Page 7, Paragraph 2: H₂ is likely not produced directly via the RBOX pathway, but is a byproduct of recycling reducing equivalents Fd_{red} and NADH that reduce H⁺ to H₂.

Page 9, Paragraph 1: Please define the criteria you are using for "High Quality" MAGs in terms of completeness and contamination. There are not 23 MAGs in Table 2, and I would not define all of them as high quality - this is ok that not all the MAGs are high quality. I suggest using <https://www.nature.com/articles/nbt.3893/tables/1>

Page 9: How was taxonomy assigned? I am certain I will find this in the methods, but I suggest bringing it up here as well. MAG taxonomy from CheckM - which is based on NCBI - will vary a lot compared to the genome taxonomy database. What I suggest is a modification to Table 2 that includes both the CheckM and GTDB assigned taxonomy along with a MAG identifier that is separate from the assigned taxonomy.

Table 2 Caption: How did the authors decide if a MAG was for the same organism? dRep is a great tool for this and it can allow the "best" MAG to be selected based on multiple criteria - weighting of completeness, contamination, number of scaffolds, etc. The authors should seriously consider using dRep or a similar dereplication tool to select the "best" MAGs.

Figure 4: The caption suggests normalization to number of reads per sample, which makes sense, but were these also normalized to the size of the MAG? Also, then, what are the units for the heat map intensity? I suggest using a standard normalization approach to account for differences in reads per sample and MAG size. There are a few ways to do this and each has its own limitations and utility. One approach will allow you to estimate the relative abundance of each by adjusting for total reads mapped and genome size. (If this an open review and you see my name, feel free to e-mail me-if not, there has been a lot of discussion of this topic on bioinformatics discussion forums).

Page 17: For 16S rRNA gene amplicon sequencing, what database was used for assigning taxonomy? Also, what regions of the 16S sequence were targeted-V3/V4?

Page 18: For the metagenomic analyses, I strongly suggest binning the contigs with additional binning platforms - e.g., MaxBin2, MetaBat2, CONCOCT then using a dereplication tool (e.g., dRep) to select the "best" bins based on some combination of completeness, contamination, and number of scaffolds. This is very likely to result in higher quality MAGs. I also strongly suggest that the authors assign taxonomy using GTDB as well as the default NCBI. GTDB provides much better taxonomic classification for MAGs from these kinds of environments. GTDB-tk can be used for this purpose: <https://gtdb.ecogenomic.org/> In the last metagenomics experiment I performed, I had 55 MAGs that were only assigned at genus level or higher with CheckM/NCBI; when I used GTDB, I ended up with only 4 that were unassigned at the species level.

Reviewer #2 (Comments for the Author):

Summary

The study used ethanol as an electron donor to examine the microbiota and metabolic pathways that produce medium-chain carboxylates, especially n-caprylate, which is less known compared with caproate. The research was conducted in reactors with in-line product extraction and aimed to shed light on the microbiota associated with n-caprylate production in open-culture

reactors. The finding of confirming *Oscillibacter* was the n-caprylate producer is clear and valuable.

Overall, the paper is well-written and contains interesting results that merit publication. However, to improve readability and clarity, some points need further clarification and certain statements require additional justification. As for the experiment and the paper, I have some suggestions which I believe can help enhance the study's quality.

Specific comments:

1. A 75-day startup period was mentioned. Please provide the relevant experimental data to describe the performance of this period.
2. What are the considerations in setting an organic loading rate of 1.5×10^2 {plus minus} 4.6 mM C L⁻¹ d⁻¹? What was the SRT during the experiment?
3. What are the basic physicochemical properties of the inoculum? Please describe it.
4. Minor differences in organic loading rates were applied. Please explain the reason for the different gradient settings.
5. During Period 2, gas sparging of N₂ gas was tested out (i.e., gas sparging was off and on between Days 143 to 184). Did it have a regular frequency?
6. Is there stirring during the reaction? What are the parameters?
7. There was some confusion in the description of the samples for shotgun metagenomic analysis. Please change the expression. And explain the reason why using the pooled sample.
8. In the reactor R3, there was no gas during period 1 but it has a gas flow rate in Table 1. Please describe how to get this data and what the gas was.
9. How did the ethanol added in each reactor behave? What was the concentration of ethanol added and the ratio to acetic acid?
10. The reactor tightness and material diffusiveness influence the H₂ partial pressures. Please explain if this has been verified or if it is just speculation.
11. Were oxygen levels tested throughout the experiment? How did the exact values change?
12. The effect of H₂ on n-caprylate production was not uniform in all reactors. The productivity decreased when the amount of H₂ decreased. Since the results of the reactor 1 and reactor 2 were more similar, why did they behave differently when the gas was passed through them?
13. Which metagenomes in reactors had a complete or nearly complete RBOX pathway was investigated. Some studies have shown that the fatty acid biosynthesis pathway may play a role as well. Has this been considered during the data analysis process? Although as you pointed out, FAB is a common pathway used by all bacteria to build their phospholipid membranes, I still suggest to add a similar analysis like Figure 5 for RBOX pathway.
14. The abbreviation RBOX should be defined for the first time.
15. For Reactors 1 and 2, *Pseudoclavibacter caeni* was an abundant bacterium. But it has no obvious relation to the n-caprylate generation. What role does it play in the microbial community? Could you please describe more about its metabolism?
16. Ruminococcaceae bacterium D5, which was only found for Reactor 3 during Periods 3B and 3C had the complete RBOX pathway in its metagenome but not in its proteome. Was it involved in the reaction and what role does it play? Please explain it.
17. The title of the article is "The gases H₂ and O₂ in open-culture reactors influence the performance and microbiota of chain elongation into n-caproate and n-caprylate". However, in the manuscript, the mechanism by which H₂ and O₂ affect reactor performance and microbial communities was not very clearly explained in the article. The focus was on the discovery of a community of organisms that could produce n-caprylate through the analysis of biological data. It is therefore recommended that the title of the article be changed accordingly to match the content of the article.
18. It is suggested to highlight important findings and include the highlights of this work.
19. What is the specific practical significance of the research? How does it guide the process in reality?
20. *Methanobrevibacter* has high abundance. Could you please explain it?
21. There were some minor issues with the references, including incomplete citations. Please check the references carefully and make the necessary corrections.
22. There are still several misleading grammatical errors and improper statements in the present version of the manuscript, please check it prudentially.

Summary

The study used ethanol as an electron donor to examine the microbiota and metabolic pathways that produce medium-chain carboxylates, especially n-caprylate, which is less known compared with caproate. The research was conducted in reactors with in-line product extraction and aimed to shed light on the microbiota associated with n-caprylate production in open-culture reactors. The finding of confirming *Oscillibacter* as the n-caprylate producer is clear and valuable.

Overall, the paper is well-written and contains interesting results that merit publication. However, to improve readability and clarity, some points need further clarification and certain statements require additional justification. As for the experiment and the paper, I have some suggestions which I believe can help enhance the study's quality.

Specific comments:

1. A 75-day startup period was mentioned. Please provide the relevant experimental data to describe the performance of this period.
2. What are the considerations in setting an organic loading rate of $1.5 \times 10^2 \pm 4.6$ mM C L⁻¹ d⁻¹? What was the SRT during the experiment?
3. What are the basic physicochemical properties of the inoculum? Please describe it.
4. Minor differences in organic loading rates were applied. Please explain the reason for the different gradient settings.
5. During Period 2, gas sparging of N₂ gas was tested out (i.e., gas sparging was off and on between Days 143 to 184). Did it have a regular frequency?
6. Is there stirring during the reaction? What are the parameters?
7. There was some confusion in the description of the samples for shotgun metagenomic analysis. Please change the expression. And explain the reason why using the pooled sample.
8. In the reactor R3, there was no gas during period 1 but it has a gas flow rate in Table 1. Please describe how to get this data and what the gas was.
9. How did the ethanol added in each reactor behave? What was the concentration of ethanol added and the ratio to acetic acid?

10. The reactor tightness and material diffusiveness influence the H₂ partial pressures. Please explain if this has been verified or if it is just speculation.
11. Were oxygen levels tested throughout the experiment? How did the exact values change?
12. The effect of H₂ on n-caprylate production was not uniform in all reactors. The productivity decreased when the amount of H₂ decreased. Since the results of the reactor 1 and reactor 2 were more similar, why did they behave differently when the gas was passed through them?
13. Which metagenomes in reactors had a complete or nearly complete RBOX pathway was investigated. Some studies have shown that the fatty acid biosynthesis pathway may play a role as well. Has this been considered during the data analysis process? Although as you pointed out, FAB is a common pathway used by all bacteria to build their phospholipid membranes, I still suggest to add a similar analysis like Figure 5 for RBOX pathway.
14. The abbreviation RBOX should be defined for the first time.
15. For Reactors 1 and 2, *Pseudoclavibacter caeni* was an abundant bacterium. But it has no obvious relation to the n-caprylate generation. What role does it play in the microbial community? Could you please describe more about its metabolism?
16. Ruminococcaceae bacterium D5, which was only found for Reactor 3 during Periods 3B and 3C had the complete RBOX pathway in its metagenome but not in its proteome. Was it involved in the reaction and what role does it play? Please explain it.
17. The title of the article is “The gases H₂ and O₂ in open-culture reactors influence the performance and microbiota of chain elongation into n-caproate and n-caprylate”. However, in the manuscript, the mechanism by which H₂ and O₂ affect reactor performance and microbial communities was not very clearly explained in the article. The focus was on the discovery of a community of organisms that could produce n-caprylate through the analysis of biological data. It is therefore recommended that the title of the article be changed accordingly to match the content of the article.
18. It is suggested to highlight important findings and include the highlights of this work.
19. What is the specific practical significance of the research? How does it guide the process in reality?
20. Methanobrevibacter has high abundance. Could you please explain it?

21. There were some minor issues with the references, including incomplete citations. Please check the references carefully and make the necessary corrections.
22. There are still several misleading grammatical errors and improper statements in the present version of the manuscript, please check it prudentially.

Dear Editor,

We are pleased to submit revisions to our manuscript entitled “Variability in *n*-caprylate and *n*-caproate producing microbiomes in reactors with in-line product extraction.” to be considered for publication as a research article in mSystems (we had changed the title based on a reviewer request).

We thank the reviewer for her/his review. On the following pages, we have addressed each comment individually in **blue font**. The changes made to the manuscript are highlighted in the revised manuscript by using the editor function. In addition, for some comments, in **red font**, we copy the sentence(s) that we added. For the edited version of the manuscript, we switched off the Track Changes for the change of figures and for reformatting the bibliography.

Answer to Reviewers' comments:

Reviewer #1 (Comments for the Author):

In this manuscript by Spirito, et al., the authors use metagenomics and proteomics to assess the impact of oxygen and hydrogen on a "chain elongation" bioreactor. Chain elongation has emerged as an area of significant research interest to convert organic wastes into beneficial products, however strategies to control microbiome behavior - especially controlling the fermentation end products -- remain elusive. Open-culture chain elongation is also a fantastic model system for understanding conversion processes in anaerobic microbiomes. Therefore, this manuscript should be of interest to readers of mSystems.

Within the specific field of chain elongation, the authors seek to answer an important question on the actual impacts of hydrogen and oxygen gas on the chain elongation process. The first is important because hydrogen is known to be create thermodynamic bottlenecks in anaerobic processes, but it has been suggested that hydrogen accumulation can be a positive in chain elongation by inhibiting competing biochemical transformations. Further, hydrogen has been proposed as an electron donor to improve chain elongation, but the evidence on the role of hydrogen is mixed thus far. Less interest has been paid to oxygen, but I have personally observed that microbial communities flanking chain elongators often include oxygen-tolerant species. Given that these are open-culture systems with complex feedstocks that are not pre-reduced, oxygen likely impacts the microbial communities. Therefore, I think that is an important topic for the field of chain elongation.

The paper is well-written, the analyses are well described, and the conclusions are justified. My comments relate mostly to the metagenomic approaches employed which can be improved. I provide point-by-point comments below.

Thank you we appreciate your feedback. We have addressed your comments below.

Introduction, Paragraph 2: Please check this first sentence. There is not any substrate-level phosphorylation occurring by acetaldehyde

dehydrogenase or alcohol dehydrogenase. Also, ATP may be produced through substrate level phosphorylation with acetate kinase. ATP is not needed to run RBOX-actually, the model you show later on (Figure 1) is great, so if the description matches these, it'll be perfect.

Thank you, we have edited the text:

“Medium-chain carboxylates are often produced *via* the reverse β -oxidation (RBOX) pathway in which ethanol, lactic acid, or another electron donor is oxidized to acetyl-CoA. **Short-chain carboxylates, such as acetate and *n*-butyrate, are then chain elongated to longer-chain carboxylates, such as *n*-caproate (six-carbon chain) and *n*-caprylate (eight-carbon chain) (1, 19-21) (Fig. 1).”**

Introduction, Paragraph 2: Your interpretation of the FAB pathway is perfectly correct and I wish it could be bolded or highlighted and shared widely.

Thank you for your comment. We agree with this reviewer.

Results, Page 7, Paragraph 2: H₂ is likely not produced directly via the RBOX pathway, but is a byproduct of recycling reducing equivalents Fd_{red} and NADH that reduce H⁺ to H₂.

Thank you, we have edited the text:

The reducing equivalents Fd_{red} and NADH produced by the RBOX pathway can reduce the H⁺ produced by the pathway to H₂ (Fig. 1).

Page 9, Paragraph 1: Please define the criteria you are using for "High Quality" MAGs in terms of completeness and contamination. There are not 23 MAGs in Table 2, and I would not define all of them as high quality - this is ok that not all the MAGs are high quality. I suggest using <https://www.nature.com/articles/nbt.3893/tables/1d>

Thank you, we edited the text to change the criteria for “High Quality” MAGs according to the provided literature.

We assembled 32 draft genomes from this data, 25 of which were high-quality (>90% completion, <5% contamination), as detailed in Table 2.

Page 9: How was taxonomy assigned? I am certain I will find this in the methods, but I suggest bringing it up here as well. MAG taxonomy from CheckM - which is based on NCBI - will vary a lot compared to the genome taxonomy database. What I suggest is a modification to Table 2 that includes both the CheckM and GTDB assigned taxonomy along with a MAG identifier that is separate from the assigned taxonomy.

Thank you. Originally we assigned the taxonomy using DIAMOND+MEGAN and the NCBI-Nr database. We redid the analysis, now providing both the taxonomic assignments from GTDB-tk (r220) and NCBI. We have added clarification to the text (in the Table 2 caption & in the methods section):

“Table 2. Metagenome-assembled genomes (MAGs) found in the reactors. The table depicts a MAG identifier, the taxonomy of the bins assigned by GTDB-tk, and the corresponding NCBI name. Species names are displayed. If no species classification was found by GTDB-tk, the genus name is displayed. The completeness and contamination computed by CheckM, and the corresponding period in the reactor are also displayed.”

“The best dereplicated bins were then checked for quality using CheckM (59) and taxonomy was assigned using GTDB-tk (release 220).”

Table 2 Caption: How did the authors decide if a MAG was for the same organism? dRep is a great tool for this and it can allow the "best" MAG to be selected based on multiple criteria - weighting of completeness, contamination, number of scaffolds, etc. The authors should seriously consider using dRep or a similar dereplication tool to select the "best" MAGs.

Thank you. We did not use dereplication previously. Instead we selected the MAGs based purely on completeness and contamination. We now changed the analysis pipeline to include dRep after binning the metagenome contigs using metabat2. We now use all contigs as input for dereplication with dRep and then continued analyzing only the dereplicated metagenome bins. We included this in the text.

The resulting bins from all samples were then dereplicated using dRep2.

Figure 4: The caption suggests normalization to number of reads per sample, which makes sense, but were these also normalized to the size of the MAG? Also, then, what are the units for the heat map intensity? I suggest using a standard normalization approach to account for differences in reads per sample and MAG size. There are a few ways to do this and each has its own limitations and utility. One approach will allow you to estimate the relative abundance of each by adjusting for total reads mapped and genome size. (If this an open review and you see my name, feel free to e-mail me-if not, there has been a lot of discussion of this topic on bioinformatics discussion forums).

Thank you for the feedback. We did not normalize for MAG size as our taxonomic analysis was not based on the MAGs we assembled. Instead we used DIAMOND to align the raw sequencing reads to the NCBI-nr protein database and then normalized for sample size using MEGAN6. This is clarified in the following part included in our methods section.

Trimmed reads were aligned to NCBI-nr database (Feb 2021) using DIAMOND (56) version 2.0.7.in blastx mode. The following parameters were used: --outfmt 100 -c1 -b12 -p 32 --top 10 -e 0.001. Resulting alignments were meganized for further analysis using daa-meganizer, which is a tool that is included in MEGAN6 (57). DIAMOND output files were loaded into MEGAN6, were normalized by sample size, and read counts were extracted for each MAG. Heatmaps were created using a Python script, only displaying MAGs with more than 12k aligned reads (Fig. 4).

Page 17: For 16S rRNA gene amplicon sequencing, what database was used for assigning taxonomy? Also, what regions of the 16S sequence were targeted-V3/V4?

We have added text to indicate which region of the 16S rRNA gene sequence was targeted:

PCR amplification with 515-forward and 806-reverse Golay barcoded primers targeting the V4 region of the 16S rRNA gene of the extracted DNA was described previously (51)

We analyzed the resulting 16S rRNA gene sequencing reads using QIIME 2 2017.3 (52) and the Silva database release 138.1.

Page 18: For the metagenomic analyses, I strongly suggest binning the contigs with additional binning platforms - e.g., MaxBin2, MetaBat2, CONCOCT then using a dereplication tool (e.g., dRep) to select the "best" bins based on some combination of completeness, contamination, and number of scaffolds. This is very likely to result in higher quality MAGs. I also strongly suggest that the authors assign taxonomy using GTDB as well as the default NCBI. GTDB provides much better taxonomic classification for MAGs from these kinds of environments. GTDB-tk can be used for this purpose: <https://gtdb.ecogenomic.org/> In the last metagenomics experiment I performed, I had 55 MAGs that were only assigned at genus level or higher with CheckM/NCBI; when I used GTDB, I ended up with only 4 that were unassigned at the species level.

Thank you very much. We changed our pipeline according to your suggestions. We performed binning using maxbin2 and then dereplicated the bins from all samples using dRep2. The taxonomy of the dereplicated bins was then assessed using gtdb-tk (r220). If possible we provide species names, for some bins only the genus or class name could be found using gtdb-tk. We also provide the taxonomic annotations from NCBI together with a MAG identifier.

The text in the methods has been updated:

De-novo Assembly

We performed de novo assembly for each set of quality filtered reads using MEGAHIT (58) version 1.2.9 with preset meta-large for large and complex metagenomes. This resulted in 757,643 contigs with a mean length of 1129 bp and a mean N50 of 2620 bp. Assembled contigs were binned using MetaBAT_2 with default parameters. The resulting bins from all samples were then dereplicated using dRep2.

The text in the results section has been updated:

All bacteria with the complete RBOX pathway in their metagenome and proteome were in the class *Clostridia*, except for *Azospira inquinata* and JAAYAE01 (NCBI

classification: *Acholeplasmataceae* bacterium) (Fig. 5). In the 16S rRNA gene sequencing data, an OTU classified as the *Clostridia* class member *O. valericigenes* dominated Reactor 1 during periods of high *n*-caprylate production (Fig. 3). A MAG classified as *Oscillibacter* contained all the RBOX enzymes in its metagenome and proteome (Fig. 5). The other bacteria in the class *Clostridia* which had the complete RBOX pathway were not dominant bacteria: *Clostridium AV fermenticellae*, *Vescimonas*, *Desulfitobacterium*, *Clostridium AM*, *Caproicibacterium* sp002411615, and *Fimivivens* (Fig. 5). In a *prior* study, which utilized inoculum from this study's reactors, a *Caproiciproducens* strain (7D4C2) was isolated from reactor biomass and shown to produce *n*-caproate (39). In our reactors, the metagenome of *Clostridium B* (NCBI: *C. kluyveri* DSM 555 (*Clostridium B*, Fig. 5, Table 2) contained and expressed the majority of RBOX enzymes except for acetyl CoA-transferase (CoAT) was not found in the proteome and electron-transfer-flavoprotein subunit B (EtfB) was not found in the proteome or metagenome (Fig. 5). It is important to note that the absence of a protein in our comparative proteomics does not mean that the protein is not present.

The text in the discussion has been updated:

As expected, *C. kluyveri* (*Clostridium B*) had most of the RBOX pathway in the metagenome and proteome (Fig. 5). Some *Clostridia* class members had the complete RBOX pathway in their metagenome and proteome, including an *Oscillibacter* species (Fig. 5). We also note that some bacteria found in our study, specifically *Azospira inquinata* and JAAYAE01 (*Acholeplasmataceae* bacterium) had the complete RBOX pathway (Fig. 5), but are not known chain elongators. Previous researchers have also noted the presence of the RBOX pathway in bacteria that are not known chain elongators (39).

Reviewer #2 (Comments for the Author):

Summary

The study used ethanol as an electron donor to examine the microbiota and metabolic pathways that produce medium-chain carboxylates, especially *n*-caprylate, which is less known compared with caproate. The research was conducted in reactors with in-line product extraction and aimed to shed light on the microbiota associated with *n*-caprylate production in open-culture reactors. The finding of confirming *Oscillibacter* was the *n*-caprylate producer is clear and valuable.

Overall, the paper is well-written and contains interesting results that merit publication. However, to improve readability and clarity, some points need further clarification and certain statements require additional justification. As for the experiment and the paper, I have some suggestions which I believe can help enhance the study's quality.

Thank you for your thoughtful feedback. We have addressed your specific comments below.

Specific comments:

1. A 75-day startup period was mentioned. Please provide the relevant experimental data to describe the performance of this period.

We have added text in the results section to highlight where the experimental data from the startup period can be found in the supplementary materials. We note that we did not calculate the *n*-caprylate or *n*-caproate production rates during the startup.

The effluent carboxylate and ethanol concentrations during the 75-day reactor startup period can be found in **Fig. S2**.

We also added text to the methods section to clarify what was done during the startup period.

During the 75-day startup period, the organic loading rates to the reactors were incrementally increased. After the startup period, we mixed broth from all reactors to ensure similar microbiota in each reactor before a restart.

2. What are the considerations in setting an organic loading rate of 1.5×10^2 {plus minus} $4.6 \text{ mM C L}^{-1} \text{ d}^{-1}$? What was the SRT during the experiment?

In a prior study in our lab (citation included below here), we observed that lower organic loading rates improved the stability and performance of the bioreactor system. We note that the organic loading rates used in our current study were lower than this prior study. However, different bioreactor designs were used in our current study, as compared to this study. This study used a continuously stirred bioreactor, whereas the prior study used an upflow anaerobic filter. We expect that the solids retention time in the prior study was longer than in this study, though this was not measured directly in either study.

We also note that the main goal of this current study was to gain an understanding of the microbiome involved in chain elongation rather than to optimize the production of *n*-caprylate & *n*-caproate.

Spirito CM, Marzilli AM, Angenent LT. 2018. Higher substrate ratios of ethanol to acetate steered chain elongation toward *n*-caprylate in a bioreactor with product extraction. *Environmental Science & Technology* 52:13438-13447.

Changes in the text are shown below:

We operated the reactors as replicates in which we kept organic loading rates and hydraulic retention time at $1.5 \times 10^2 \pm 4.6 \text{ mM C L}^{-1} \text{ d}^{-1}$ and 8.5 ± 0.2 days, respectively, for a period of 68 days (Period 1 of study – Days 75 to 142; see **Table 1**). **This organic loading rate was lower than in prior studies in our lab with ethanol and acetate-fed bioreactors (3,4). Still, the different reactor designs should be noted (*i.e.*, continuously mixed bioreactors in this study vs. upflow anaerobic filters in the prior studies). The solids retention time was not measured in our bioreactors.**

The text is referring to these two papers:

Kucek LA, Spirito CM, Angenent LT. 2016. High *n*-caprylate productivities and specificities from dilute ethanol and acetate: Chain elongation with microbiomes to upgrade products from syngas fermentation. *Energy & Environmental Science* 9:3482-3494.

Spirito CM, Marzilli AM, Angenent LT. 2018. Higher substrate ratios of ethanol to acetate steered chain elongation toward *n*-caprylate in a bioreactor with product extraction. *Environmental Science & Technology* 52:13438-13447.

3. What are the basic physicochemical properties of the inoculum? Please describe it.

We have not performed physiochemical characterization of the inoculum for this study. The physiochemical characteristics of the bioreactor from which the inoculum came from are reported in the following two studies that we cite in the text:

Agler MT, Spirito CM, Usack JG, Werner JJ, Angenent LT. 2012. Chain elongation with reactor microbiomes: Upgrading dilute ethanol to medium-chain carboxylates. *Energy and Environ Science* 5:8189-8192.

Ge S, Usack JG, Spirito CM, Angenent LT. 2015. Long-term *n*-caproic acid production from yeast-fermentation beer in an anaerobic bioreactor with continuous product extraction. *Environmental Science and Technology* 49:8012-8021.

We have modified the main text to include details on how this prior reactor was fed:

Reactors were inoculated with 10% by volume (~500 mL) of reactor broth from a reactor that was fed **semi-continuously (~once every two days)** with ethanol-rich yeast fermentation beer and operated as an anaerobic sequencing batch reactor for an operating period of approximately five years *prior* to the time we collected the inoculum (7, 16).

We realized this above text was duplicated in our supplementary information, so we have removed it from the supplementary information.

4. Minor differences in organic loading rates were applied. Please explain the reason for the different gradient settings.

We aimed for similar organic loading rates to all the bioreactors in the study. However, small differences in influent flow rates and the composition of the media substrate contributed to the minor differences in the organic loading rates.

We have added text to note this in the main text (in the methods section):

Throughout Periods 1 to 3, we aimed for similar organic loading rates to all bioreactors. Relatively small differences in organic loading rates (Table 1) can be attributed to minor

differences in the influent flow rate and prepared influent composition supplied to the three bioreactors.

5. During Period 2, gas sparging of N₂ gas was tested out (i.e., gas sparging was off and on between Days 143 to 184). Did it have a regular frequency?

There was not a consistent pattern to the testing of the gas sparging during Period 2. We have updated the text to note this:

During Period 2, gas sparging of N₂ gas was tested out (i.e., gas sparging was off and on **irregularly** between Days 143 to 184) (**Table 1**).

6. Is there stirring during the reaction? What are the parameters?

We have added details on how the reactors were continuously mixed:

The reactors were continuously mixed *via* a peristaltic pump (Cole Parmer, Part No. 7520-10), which recirculated the reactor broth at a rate of ~40 mL min⁻¹ by removing broth from the top of the reactor liquid level and returning it to the bioreactor base (internal recycle line; **Fig. S1**). We **continuously** fed the reactors with a modified-based media that was previously described (4, 48) and supplemented with ethanol and acetate.

7. There was some confusion in the description of the samples for shotgun metagenomic analysis. Please change the expression. And explain the reason why using the pooled sample.

Thank you for your comments. We have addressed them in the text:

We collected biomass samples for shotgun metagenomic analysis **approximately weekly** from internal liquid-recycle lines of the reactors, which were utilized to mix the reactor liquid. Samples were centrifuged, supernatant was discarded, and biomass was stored at -80°C. Genomic DNA was extracted using the PowerSoil DNA Isolation kit (MO BIO Laboratories Inc., Carlsbad, CA). We used a modified protocol, which has been described by Kucek et al. (4). **After quantifying the extracted DNA, we selected nine samples for shotgun metagenomics sequencing (three samples for each reactor during Periods 1C, 3B, and 3C). For Period 1C, we selected one sample from Reactors 1 and 2 on Day 137 and a pooled sample from Reactor 3 from Days 137, 151, 154, and 162. For Period 3B, we selected a pooled sample from each reactor on Days 198 and 200. For Period 3C, we selected one sample from each reactor on Day 218. Pooled samples were utilized if the concentration of the genomic DNA extracted was low on a single day. The nine selected DNA samples were barcoded and sequenced on two lanes (100 bp *per* read; single-direction reads) using Illumina HiSeq platform at the JP Sulzberger Genome Center at Columbia University (New York, New York). We merged the replicates of samples.**

8. In the reactor R3, there was no gas during period 1 but it has a gas flow rate in Table 1. Please describe how to get this data and what the gas was.

The reviewer is correct in noting that no external gas was added to any of the reactors during Period 1, as noted in Table 1. We suspect the gas flow measured in Reactor 3

during this period is due to the production of hydrogen gas indirectly via the reverse beta oxidation pathway. We note the higher percentage of hydrogen measured in the headspace of Reactor 3 during this period (compared to the other reactors).

The authors direct the reviewer to this section of the main text (in the results):

Our results show that the H₂ partial pressure is a sensitive parameter to the *n*-caprylate performance, amplifying minor differences in operating conditions. During Period 1, gas in the headspace of Reactor 3 contained 31 ± 9.6% H₂ (by volume), whereas H₂ was 9.9 ± 5.2% and 1.8 ± 1.9% of total gas for Reactors 1 and 2, respectively (**Table S1**). The reducing equivalents Fd_{red} and NADH produced by the RBOX pathway can reduce the H⁺ produced by the pathway to H₂ (**Fig. 1**). The reactor tightness and material diffusiveness may influence the H₂ partial pressures because H₂, as the smallest molecule, may easily diffuse out of the system, while other gases would not. We built almost the entire reactor setup out of stainless steel to minimize H₂ diffusion through plastic tubing and connections. However, our results show that we were not able to prevent H₂ diffusion out of the system, which included a gas recirculation pump and some tubing lines that were not made of stainless steel.

The method used to determine the composition of the headspace gas can be found in the main text methods:

We collected gas samples from gas exit lines of the reactors. CO₂, CH₄, and H₂ concentrations (>0.2% by volume) were measured using a gas chromatography system, which has been described previously (49). A reduction gas detector (RGD) was used to measure H₂ gas concentrations <0.2%, which has been described by Kucek et al. (4).

The method used to measure the gas flow rate can be found in the supporting information:

Gas exit lines from the top of the reactor led to a condensation trap, bubbler, and then a gas flow meter (Calibrated Instruments Inc., Ritter MilliGas Counter Series MGC-1 V3.1, Hawthorne, NY).

9. How did the ethanol added in each reactor behave? What was the concentration of ethanol added and the ratio to acetic acid?

Thank you for your feedback. We have added a sentence to the methods to report on the ethanol and acetate ratios/concentrations in the media:

During Periods 1 to 3 of the study, the molar ratio of ethanol to acetate was maintained at 10:1 in the substrate, and the ethanol concentration was ~600 mM.

We also direct the reviewer to existing text in the results on the ethanol concentrations in the reactors. Please see Figure S2 and Table S2 for the effluent ethanol concentrations in the reactors.

With the relatively high H₂ partial pressures for Reactor 3 during Period 1 compared to Reactors 1 and 2, a significant fraction of ethanol that we fed to Reactor 3 was not converted and left in the effluent, which resulted in a higher average effluent ethanol concentration for Reactor 3 compared to Reactor 1 and 2 ($1.7 \times 10^2 \pm 9.7$ mM vs. 47 ± 3.9 mM and 29 ± 4.3 mM, respectively (**Fig. S2, Table S2**).

10. The reactor tightness and material diffusiveness influence the H₂ partial pressures. Please explain if this has been verified or if it is just speculation.

Thank you for your comments. The reactor tightness and material diffusiveness were not tested directly. We have noted this in the text:

The reactor tightness and material diffusiveness may influence the H₂ partial pressures because H₂, as the smallest molecule, may easily diffuse out of the system, while other gases would not. We built almost the entire reactor setup out of stainless steel to minimize H₂ diffusion through plastic tubing and connections. However, our results show that we were not able to prevent H₂ diffusion out of the system, which included a gas recirculation pump and some tubing lines that were not made of stainless steel. **We note that we did not directly measure the reactor tightness and material diffusivity in this study.**

11. Were oxygen levels tested throughout the experiment? How did the exact values change?

We did not take measurements of oxygen levels in the reactors. This is a good idea for future experiments.

12. The effect of H₂ on *n*-caprylate production was not uniform in all reactors. The productivity decreased when the amount of H₂ decreased. Since the results of the reactor 1 and reactor 2 were more similar, why did they behave differently when the gas was passed through them?

The effect of H₂ on *n*-caprylate production needs further study and is not fully answered by our current study. We direct the reviewer to the section of the paper where we discuss H₂ partial pressures and their possible effects on *n*-caprylate production:

To test whether a lower H₂ partial pressure would improve *n*-caprylate production rates, we sparged N₂ gas into Reactor 3 to reduce the percentage of H₂ in the headspace (**Table 1; Table S1**). The sparging decreased the H₂ in the headspace from $31 \pm 9.6\%$ (by volume) during Period 1 to $20 \pm 14\%$ during Period 2 to $7.3 \pm 4.6\%$ during Period 3 (**Table S1**), resulting in increased volumetric *n*-caprylate production rates for Reactor 3 during Periods 2 and 3 (**Fig. 2B**). Into Reactor 2, we sparged N₂ and H₂ gas into the reactor. As expected, when hydrogen partial pressures increased during Periods 2 and 3 (**Table S1**), *n*-caprylate productivity decreased for Reactor 2 (**Figure 2B**). However, we observed that the effect of H₂ on *n*-caprylate production was not uniform in all reactors. When the amount of H₂ in the headspace decreased due to N₂ sparging into Reactor 1, we observed decreased *n*-caprylate production rates during Periods 2 and 3 (**Fig. 2B; Table**

S1). However, sparing with N₂ to remove H₂ may have also removed O₂, which could have an unknown effect. Gas sparging itself was another introduced variable in the experiment that may have decreased biomass growth and *n*-caprylate production for Reactor 1 during Periods 2 and 3. We also noted differences in the acetate, *n*-butyrate, *n*-caproate, and *n*-caprylate concentrations in the effluent of our reactors (**Table S2, Fig. S2A-C**). Thus, our system was not predictive because we did not fully understand how the environmental conditions in the reactor affect the microbial pathways in the complex microbiota.

13. Which metagenomes in reactors had a complete or nearly complete RBOX pathway was investigated. Some studies have shown that the fatty acid biosynthesis pathway may play a role as well. Has this been considered during the data analysis process? Although as you pointed out, FAB is a common pathway used by all bacteria to build their phospholipid membranes, I still suggest to add a similar analysis like Figure 5 for RBOX pathway.

Thank you for your suggestion. We did perform this analysis (of the FAB pathway) but decided not to include the results in our paper. We decided not to include the FAB pathway results because FAB is a common pathway bacteria use to build their phospholipid membranes. In addition, we did not find any correlations between the presence of the FAB pathway and *n*-caprylate production rates.

14. The abbreviation RBOX should be defined for the first time.

We define this abbreviation the first time we mention it in the text (in the introduction). We have copied the text here:

“Medium-chain carboxylates are often produced *via* the reverse β -oxidation (RBOX) pathway in which ethanol, lactic acid, or another electron donor is oxidized to acetyl-CoA”

15. For Reactors 1 and 2, *Pseudoclavibacter caeni* was an abundant bacterium. But it has no obvious relation to the *n*-caprylate generation. What role does it play in the microbial community? Could you please describe more about its metabolism?

The metabolism of *Pseudoclavibacter caeni* and its role in the chain elongation community is not entirely known and is a focus of ongoing research. We agree with the reviewer that this is an interesting question. We direct the reviewer to the portion of the paper (the discussion) where we discuss the aerobic *P. caeni*'s possible role as an oxygen scavenger in our system:

The unplanned presence of O₂ in our reactors created a niche for aerobic bacteria, such as *P. caeni* and *Acetobacter* species, to survive and become abundant in the reactors (**Fig. 4**). The abundance of these aerobic bacteria was not correlated to *n*-caprylate production rates (**Figs. 3-4**). As a result of our inability to build a reactor system that prevented O₂ inclusion, a major caveat existed in our quest to study different H₂ partial pressures on the RBOX. The use of gas sparging to remove or add H₂, also removed O₂, and this turned out to be a sensitive parameter. Even though, we could not satisfy our experimental

design with the independent parameter H_2 , this study is providing us with information to base future research on, as discussed below. Aerobic or facultative anaerobic microbes must have quickly consumed the O_2 in our reactors because strict anaerobic microbes, such as methanogens and other obligate anaerobes, were also present in our continuously stirred reactor systems (**Fig. 4**). *Prior* studies observed aerobes, such as *Acetobacter* (3, 11), and facultative anaerobes, such as *Lactobacillus* (11, 45), in chain elongation reactors. Previous studies from our lab had not found *P. caeni* in similar chain-elongating reactors (3, 4), though, the aerobe *Acetobacter* was observed (3). *P. caeni* was isolated from sewage sludge in 2012 (46), but the *P. caeni* assembly was only added to the NCBI nr database in 2019 (ASM883112v1). *P. caeni* could have been present in previous reactor studies but not detected due to its absence from existing databases. A previous study from 2016 found a phylotype that matches *P. caeni* in batch experiments utilizing biomass from a chain-elongating reactor that was fed a variety of substrates and found its occurrence did not correspond to chain elongation activity (47).

16. Ruminococcaceae bacterium D5, which was only found for Reactor 3 during Periods 3B and 3C had the complete RBOX pathway in its metagenome but not in its proteome. Was it involved in the reaction and what role does it play? Please explain it.

We don't know the exact role of this bacterium in the reactor microbiome, though we speculate it was directly involved in medium-chain carboxylate production. Previous 16S rRNA gene sequencing studies have found members of the Ruminococcaceae family to be associated with medium-chain carboxylate production, as we noted in the text:

This finding is consistent with *prior* studies for which members of the Ruminococcaceae family (to which *Oscillibacter* belongs) were isolated from reactors producing *n*-caproate from lactate (9, 29) and Illumina 16S rRNA gene sequencing studies for which Ruminococcaceae members were associated with medium-chain carboxylate production in reactors (4, 7, 11).

17. The title of the article is "The gases H_2 and O_2 in open-culture reactors influence the performance and microbiota of chain elongation into *n*-caproate and *n*-caprylate". However, in the manuscript, the mechanism by which H_2 and O_2 affect reactor performance and microbial communities was not very clearly explained in the article. The focus was on the discovery of a community of organisms that could produce *n*-caprylate through the analysis of biological data. It is therefore recommended that the title of the article be changed accordingly to match the content of the article.

Thank you for your suggestion. We have changed the article's title to "Variability in *n*-caprylate and *n*-caproate producing microbiomes in reactors with in-line product extraction."

18. It is suggested to highlight important findings and include the highlights of this work.

We direct the reviewer to the discussion where we discuss the main findings of the work. We also highlight them here:

- We found different microbial communities were correlated with periods of high n-caproate vs n-caprylate production. We identified potential candidate n-caprylate producers in our bioreactors. For example, we found that *Oscillibacter* species members were correlated with periods of high n-caprylate production.
- We observed the relatively high abundance of aerobic bacteria *P. caeni* and *Acetobacter* species in our reactors and speculate that they could be oxygen scavengers in our reactors, allowing the anaerobic bacteria that carry out the RBOX pathway to survive.
- From our metagenomic and metaproteomics analysis, we can see that the reverse beta-oxidation pathway was active in our reactors.

19. What is the specific practical significance of the research? How does it guide the process in reality?

We direct the reviewer to the importance section of our paper. We also outline the practical significance here:

- The reverse beta-oxidation pathway can be used in open-culture reactors to upgrade organic wastes into valuable biochemicals. Knowledge of the key players in the reactor microbiome and their role in the RBOX pathway can inform the scale-up of these reactor systems for industrial applications.
- This study combines 16S rRNA gene sequencing, metagenomics, and metaproteomics data with reactor operating and performance data to explore potential n-caproate and n-caprylate producers in reactors operated with in-line product extraction.
- We observe changes in the reactor microbiota under different operating conditions (H_2 partial pressures), though more work is needed to develop a mechanistic understanding of the effect of H_2 partial pressures on these systems.

20. Methanobrevibacter has high abundance. Could you please explain it?

Methanobrevibacter was not the most abundant bacteria in our reactors (Please see Figure 4, which shows the results of our shotgun metagenomics analysis). We note that Methanobrevibacter does appear on this heatmap of abundant bacteria in the reactors. Still, it is not in high abundance relative to some of the other bacteria in our reactors. Methanobrevibacter is a hydrogenotrophic methanogen. pH stress tends to inhibit acetoclastic methanogens to a greater extent than hydrogenotrophic methanogens. See:

Qiu, S., et al. (2023). "Effect of extreme pH conditions on methanogenesis: Methanogen metabolism and community structure." Science of The Total Environment **877**: 162702.

21. There were some minor issues with the references, including incomplete citations. Please check the references carefully and make the necessary corrections.

Thank you. We have corrected issues with capitalizations and bacterial name formats in the reference list.

22. There are still several misleading grammatical errors and improper statements in the present version of the manuscript, please check it prudentially.

Thank you for your suggestion. We have edited some areas of the text to improve clarity. We also made minor edits elsewhere in the text, removing or adding commas as needed:

Page 4:

Recent studies have suggested that the fatty acid biosynthesis (FAB) pathway may also play a role (26, 27). However, it should be noted that all bacteria use the anabolic FAB pathway to build their phospholipid membranes.

Page 6:

The reverse β -oxidation (RBOX) pathway investigated in this study.

Page 10:

Based on the shotgun metagenomics data, the abundance of *C. kluyveri* was negatively correlated to *n*-caprylate production rates. However, the correlation was not significant (Fig. 4, $r = -0.49$, $p=0.18$).

Page 13:

We observed the ETU microcompartment as expected in *C. kluyveri*.

Page 14:

Using gas sparging to remove or add H_2 also removed O_2 , a sensitive parameter.

Even though we could not satisfy our experimental design with the independent parameter H_2 , this study provides information on which to base future research, as discussed below.

Page 15

We did not find RNF and ECH proteins in the proteome for this bacterium; they were found only in the metagenome.

A previous study from 2016 found a phylotype that matches *P. caeni* in batch experiments utilizing biomass from a chain-elongating reactor fed a variety of substrates and found its occurrence did not correspond to chain elongation activity (47).

Page 16

The 2-mL samples of reactor broth were collected from a port in the reactor system recycle line.

Re: mSystems00416-24R1 (Variability in *n*-caprylate and *n*-caproate producing microbiomes in reactors with in-line product extraction.)

Dear Prof. Largus T Angenent:

Your manuscript has been accepted, and I am forwarding it to the ASM production staff for publication. Your paper will first be checked to make sure all elements meet the technical requirements. ASM staff will contact you if anything needs to be revised before copyediting and production can begin. Otherwise, you will be notified when your proofs are ready to be viewed.

Sincerely,
Yu-Liang Yang

Editor
mSystems

Reviewer #1 (Comments for the Author):

I have no additional comments.